# STaMP: Sequence Transformation and Mixed Precision for Low-Precision Activation Quantization

**Marco Federici**
Qualcomm AI Research
{mfederic}@qualcomm.com

**Riccardo Del Chiaro**
Qualcomm AI Research
{rdelchia}

**Boris van Breugel**
Qualcomm AI Research
{bvanbreu}

**Paul Whatmough**
Qualcomm AI Research
{pwhatmou}

**Markus Nagel**
Qualcomm AI Research
{markusn}

## Abstract

Quantization is the key method for reducing inference latency, power and memory footprint of generative AI models. However, accuracy often degrades sharply when activations are quantized below eight bits. Recent work suggests that invertible linear transformations (e.g. rotations) can aid quantization, by reparameterizing feature channels and weights. In this paper, we propose *Sequence Transformation and Mixed Precision* (STaMP) quantization, a novel strategy that applies linear transformations along the *sequence* dimension to exploit the strong local correlation in language and visual data. By keeping a small number of tokens in each intermediate activation at higher precision, we can maintain model accuracy at lower (average) activations bit-widths. We evaluate STaMP on recent LVM and LLM architectures, demonstrating that it significantly improves low bit width activation quantization and complements established activation and weight quantization methods including recent feature transformations. The code is available at `https://github.com/Qualcomm-AI-research/stamp-quantization`.

## 1 Introduction

Modern generative models, such as large language models (LLMs) and large vision models (LVMs), achieve state-of-the-art performance in text and image generation but at the cost of massive computational and memory requirements. As the demand for scalable and efficient deployment of these models grows, both in the cloud and on edge devices where computational resources are scarce, achieving inference efficiency has become a critical area of research.

Post-Training Quantization (PTQ) of weights and activations is fundamental to enhancing inference efficiency, especially for demanding operations such as large matrix multiplications in linear layers, which dominate power consumption. However, PTQ faces substantial challenges when pushing the quantization bitwidth down to 4-bits, often due to the presence of outliers in weights and activations.

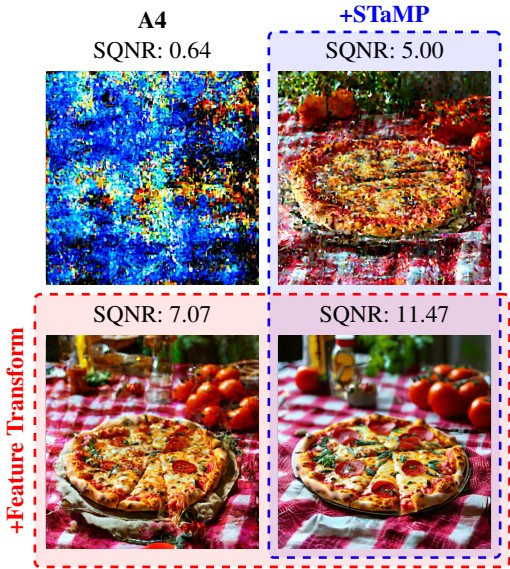

Figure 1: STaMP and feature transformations applied to PixArt-$\Sigma$ with 4-bit activations. The benefit of STaMP is orthogonal to (Hadamard) feature transformation, drastically reducing artifacts.

To mitigate outliers, prior work applies *function-preserving transformations* (van Breugel et al., 2025) to weights and activations. For example, Xiao et al. (2023) scales down outlier activations and compensates by scaling up subsequent weights, preserving the model output. Similarly, Hadamard-based feature mixing (Ashkboos et al., 2024; Liu et al., 2024; Ma et al., 2024; Zhao et al., 2025) reduces activation variance and spreads outliers across dimensions.

However, these methods operate primarily along the feature dimension and ignore correlations across the sequence dimension. Drawing inspiration from traditional media compression methods, in this paper we propose *Sequence Transformation and Mixed Precision* (STaMP) quantization, a complementary approach that leverages sequence structure to improve activation quantization.

Our main contributions are summarized below:

- We introduce a new class of activation transformations operating along the sequence dimension, complementary to existing feature transformations (Figure 1).
- We characterize the quantization error for sequence transforms and design a novel mixed-precision quantization scheme to exploit local activation correlation.
- We demonstrate that STaMP consistently improves the model accuracy when combined with feature transformations and weight quantization on both LLM and LVM.

## 2 BACKGROUND

### 2.1 ACTIVATION QUANTIZATION

Consider $\boldsymbol{X} \in \mathbb{R}^{s \times d}$ as an activation matrix of shape *sequence length $s \times$ feature size $d$* (batch size is omitted for the sake of clarity). Integer quantization refers to the operation of converting the activations to integer values with lower bit width. The activation quantization operation $\mathbf{Q} : \mathbb{R}^{s \times d} \to \mathbb{N}^{s \times d}$ is defined as:

$$x_{ij}^{\text{int}} \stackrel{def}{=} \mathbf{Q}(\boldsymbol{X})_{ij} = \text{clamp}\left(\left\lfloor \frac{x_{ij}}{s_{ij}} \right\rceil + z_{ij}, 0, 2^{b_{ij}} - 1\right), \tag{1}$$

in which $z_{ij}$ and $s_{ij}$ can be interpreted as an offset and scaling parameter, while $b_{ij}$ refers to the number of bits used to quantize the entry $x_{ij}$. In order to make quantization efficient, offsets, scales and bit widths are shared across all feature channels: $s_{ij} = s_i$, $z_{ij} = z_i$, and $b_{ij} = b_i$.

The de-quantization function $\mathbf{Q}^{-1} : \mathbb{N}^{s \times d} \to \mathbb{R}^{s \times d}$ maps the integer-quantized activation into the original domain: $\mathbf{Q}^{-1}(\boldsymbol{X}^{\text{int}})_{ij} = (x_{ij}^{\text{int}} - z_{ij})s_{ij}$. We will refer to the combination of the quantization and de-quantization operation with $\mathcal{Q}(\boldsymbol{X}) \stackrel{def}{=} \mathbf{Q}^{-1}(\mathbf{Q}(\boldsymbol{X}))$, omitting the quantization parameters for brevity. The expected activation quantization error introduced by the quantization and de-quantization operations is commonly defined as:

$$\mathcal{L}(\boldsymbol{X}) \stackrel{def}{=} \mathbb{E}\left[\|\mathcal{Q}(\boldsymbol{X}) - \boldsymbol{X}\|_2^2\right] = \sum_{i=1}^{s} \overbrace{\mathbb{E}\left[\|\mathcal{Q}(\boldsymbol{x}_i) - \boldsymbol{x}_i\|_2^2\right]}^{\mathcal{L}(\boldsymbol{x}_i)}, \tag{2}$$

in which the expectation is computed with respect to the activation distribution $p(\boldsymbol{X})$ and $\|\cdot\|_2^2$ refers to the squared Frobenius norm. Nagel et al. (2021) identified two main causes of quantization error: clipping error (due to the clamp operator) and rounding error (introduced by the rounding). As is common practice in the literature, we will focus on a setting in which the scale $s_i$ and offset $z_i$ for the i-th token are set based on the range $(\boldsymbol{x}_i) \stackrel{def}{=} \max_j \boldsymbol{x}_{ij} - \min_j \boldsymbol{x}_{ij}$ to prevent any clipping error: $\overline{s_i} \stackrel{def}{=} \frac{2^{b_i} - 1}{\text{range}(\boldsymbol{x}_i)}$, $\overline{z_i} \stackrel{def}{=} -\frac{\min_j \boldsymbol{x}_{ij}}{\overline{s_i}}$. In this setting, the quantization error for each token $\boldsymbol{x}_i$ is determined by its quantization scale $\overline{s_i}$:

$$\mathcal{L}(\boldsymbol{x}_i) \leq \frac{d}{4}\mathbb{E}\left[\overline{s_i}^2\right] = \frac{d}{4}\frac{\mathbb{E}\left[\text{range}(\boldsymbol{x}_i)^2\right]}{(2^{b_i} - 1)^2}. \tag{3}$$

As the activation quantization error increases, so does the deterioration in performance for the model outputs since the network output diverge from the original (unquantized) model.

## 2.2 FEATURE TRANSFORMATIONS

In order to reduce the activation quantization error, recent literature has introduced linear function-preserving transformations in the form of (right) invertible matrices $\boldsymbol{R}$, which are applied prior to the quantization operation (van Breugel et al., 2025). Clearly, the additional flexibility introduced by **Feature Transformations** can aid in reducing the activation quantization error:

$$\mathcal{L}\left(\boldsymbol{X}\right) \geq \min_{\boldsymbol{R}} \underbrace{\mathbb{E}\left[\left\|\mathcal{Q}\left(\boldsymbol{X}\boldsymbol{R}\right)\boldsymbol{R}^{-1} - \boldsymbol{X}\right\|_2^2\right]}_{\mathcal{L}(\boldsymbol{X};\boldsymbol{R})}. \tag{4}$$

In particular, rotation matrices can effectively reduce the activation range, effectively spreading outliers across multiple channels and consequently reducing the subsequent quantization error:

$$\sum_{i=1}^s \mathbb{E}\left[\text{range}\left(\boldsymbol{x}_i\boldsymbol{R}\right)^2\right] \leq \sum_{i=1}^s \mathbb{E}\left[\text{range}\left(\boldsymbol{x}_i\right)^2\right] \implies \mathcal{L}\left(\boldsymbol{X};\boldsymbol{R}\right) \leq \mathcal{L}\left(\boldsymbol{X}\right) \tag{5}$$

Hadamard matrices have proven very effective in reducing the number of outliers with limited additional overhead: matrix multiplications with Hadamard matrices can be performed efficiency in $O(sd\log d)$ thanks to the butterfly algorithm (Fino & Algazi, 1976) , and the inverse Hadamard matrix can be fused into linear layer weights (Ashkboos et al., 2024).

Existing feature transformation techniques reduce activation quantization error by redistributing activation ranges across features. However, they operate exclusively along the feature dimension and ignore correlations across the sequence dimension. Visual and textual data exhibit strong local correlations—neighboring pixels in images and adjacent tokens in text are strongly dependent. This suggests that a similar structure could exist in the model intermediate activations and could be leveraged to improve quantization efficiency, which is described in the next section.

## 3 METHOD

We define a **Sequence Transform** as a linear transformation of $\boldsymbol{X}$ across sequence dimension defined by a (left) invertible matrix $\boldsymbol{L}$. Similarly to feature transformations $\boldsymbol{R}$, sequence transformations can reduce the quantization error, and the two can be easily combined to further the error:

$$\mathcal{L}\left(\boldsymbol{X}\right) \geq \min_{\boldsymbol{L}} \underbrace{\mathbb{E}\left[\left\|\boldsymbol{L}^{-1}\mathcal{Q}\left(\boldsymbol{L}\boldsymbol{X}\right) - \boldsymbol{X}\right\|_2^2\right]}_{\mathcal{L}(\boldsymbol{X};\boldsymbol{L})} \geq \min_{\boldsymbol{L},\boldsymbol{R}} \underbrace{\mathbb{E}\left[\left\|\boldsymbol{L}^{-1}\mathcal{Q}\left(\boldsymbol{L}\boldsymbol{X}\boldsymbol{R}\right)\boldsymbol{R}^{-1} - \boldsymbol{X}\right\|_2^2\right]}_{\mathcal{L}(\boldsymbol{X};\boldsymbol{L},\boldsymbol{R})}. \tag{6}$$

Sequence transformations are linear, hence they commute with other linear operations. For a linear layer:

$$\left(\boldsymbol{L}^{-1}\mathcal{Q}\left(\boldsymbol{L}\boldsymbol{X}\right)\right)\boldsymbol{W} + \mathbf{1}\boldsymbol{\beta}^T = \boldsymbol{L}^{-1}\left(\mathcal{Q}\left(\boldsymbol{L}\boldsymbol{X}\right)\boldsymbol{W}\right) + \mathbf{1}\boldsymbol{\beta}^T = \boldsymbol{L}^{-1}\left(\mathcal{Q}\left(\boldsymbol{L}\boldsymbol{X}\right)\boldsymbol{W} + \underbrace{(\boldsymbol{L}\mathbf{1})}_{\boldsymbol{\ell}}\boldsymbol{\beta}^T\right), \tag{7}$$

in which $\mathbf{1}$ represents a vector of ones of size $s$, indicating that the same bias $\boldsymbol{\beta}$ is applied to all the tokens. This implies that we can invert sequence transformation (i) right before applying the bias in a linear layer, or (ii) postpone this operation and use a sequence-transformed bias $\boldsymbol{\ell}\boldsymbol{\beta}^T$ in which the scale $\ell_i$ may differ across different tokens as a function of $\boldsymbol{L}$. The algorithm for a sequence transformed linear layer is reported in Figure 2a.

We emphasize that, contrary to feature transformations, sequence transforms do not affect weights, and hence they are orthogonal to more advanced weight quantization methods such as vector quantization, GPTQ (Guo et al., 2024), and SVDQuant (Li et al., 2025).

### 3.1 SEQUENCE TRANSFORM AND MIXED PRECISION (STAMP)

To understand how sequence transformations affect quantization error, we first formalize the relationship between quantization error and sequence-transformed tokens (proof in Appendix A.1):

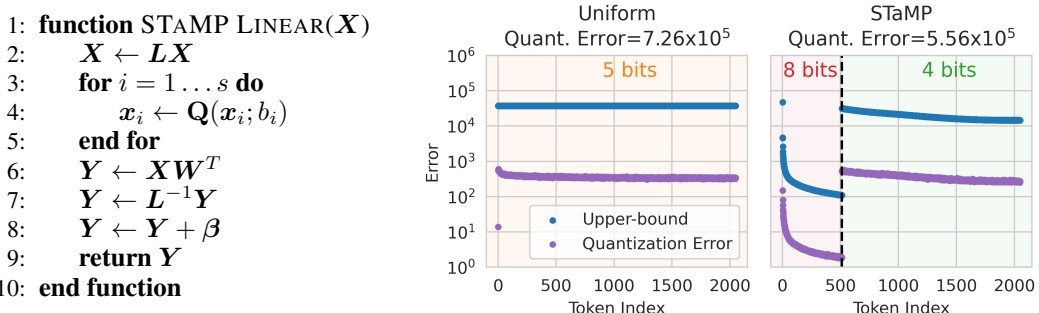

```
1: function STaMP LINEAR(X)
2:      X ← LX
3:      for i = 1 ... s do
4:          xᵢ ← Q(xᵢ; bᵢ)
5:      end for
6:      Y ← XWᵀ
7:      Y ← L⁻¹Y
8:      Y ← Y + β
9:      return Y
10: end function
```

(a) **STaMP Linear Layer Pseudocode**    (b) **Comparison of Upper-Bound and Quantization error**

Figure 2: **Summary of the STaMP Procedure**. The sequence Transform $L$ aims to concentrate the energy in the initial tokens, which are quantized at higher precision. This reduces the value of the upper-bound in Equation 8 (blue) and, consequently, the overall activation quantization error (purple). For a fixed average bit width of 5 bits, combining energy concentration with two precision levels (2b, right) achieves lower error than a uniform quantization scheme without sequence transformations (2b, left). Activations are collected from the input to Layer 20 of LLaMA v3 8B.

**Theorem 1.** *The expected quantization error for activations $X$ transformed by an orthogonal sequence transformation $L$ and quantized using a min-max scale for each token is upper-bounded by the weighted sum of the expected norm of the transformed tokens:*

$$\mathcal{L}\left(X;L\right) \le \frac{d}{2}\sum_{i=1}^{s}\frac{\overbrace{\mathbb{E}\left[\left\|l_i^T X\right\|_2^2\right]}^{e_i}}{\left(2^{b_i}-1\right)^2}. \tag{8}$$

Note that sequence transformations $L$ do not alter the *total energy* $E = \sum_{i=1}^{s} e_i$, so any improvement to the bound in Equation 8 must come from redistributing the bit width. This observation motivates a mixed precision strategy: instead of keeping energy and bit width uniform, we deliberately concentrate most of the energy into a few tokens and allocate more bits to them. Because the denominator in Equation 8 grows exponentially with $b_i$, allocating extra bits to tokens with large energy yields a disproportionately large reduction in their contribution to the error. In other words, redistributing a bit from a low-energy token to a high-energy token reduces the total error more than keeping bit widths and energy uniform. This property is illustrated in Figure 2b and further elaborated in Appendix A.3.

Therefore, to improve activation quantization performance, we propose **Sequence Transform and Mixed Precision** (STaMP) a simple strategy that concentrates activation energy into a small set of tokens and assigns them higher precision. In the next section, we describe how to design a transformation $L$ that achieves this efficiently and how to determine the bit width allocation.

## 3.2 EFFICIENT ENERGY CONCENTRATION

The energy of the i-th sequence-transformed token $e_i$ can be also seen as the projection of the autocorrelation matrix $S = \mathbb{E}[XX^T]$ along the direction $l_i$:

$$e_i = \mathbb{E}\left[\|l_i X\|_2^2\right] = l_i^T S l_i. \tag{9}$$

he direction that maximizes this energy is the eigenvector $u_1$ associated with the largest eigenvalue $\lambda_1$ of $S$. Similarly, the second largest energy corresponds to $u_2$ and so on. Therefore, given the eigendecomposition $S = U\Lambda U^T$, the optimal orthogonal transformation $L$ for concentrating the token norm is $U^T$. In this case, the energy $e$ of the transformed tokens $LX$ aligns with the squared eigenvalues $\lambda^2$. This linear transformation is also known as the **Karhunen-Loève Transform** (KLT), which requires a representative calibration set to estimate $U$. Despite its optimality, KLT has the same computational complexity of a full-rank matrix multiplication, which is impractical since each transform needs to be applied twice for each linear layer. Estimating $S$ for each activation would further adds a costly calibration step.

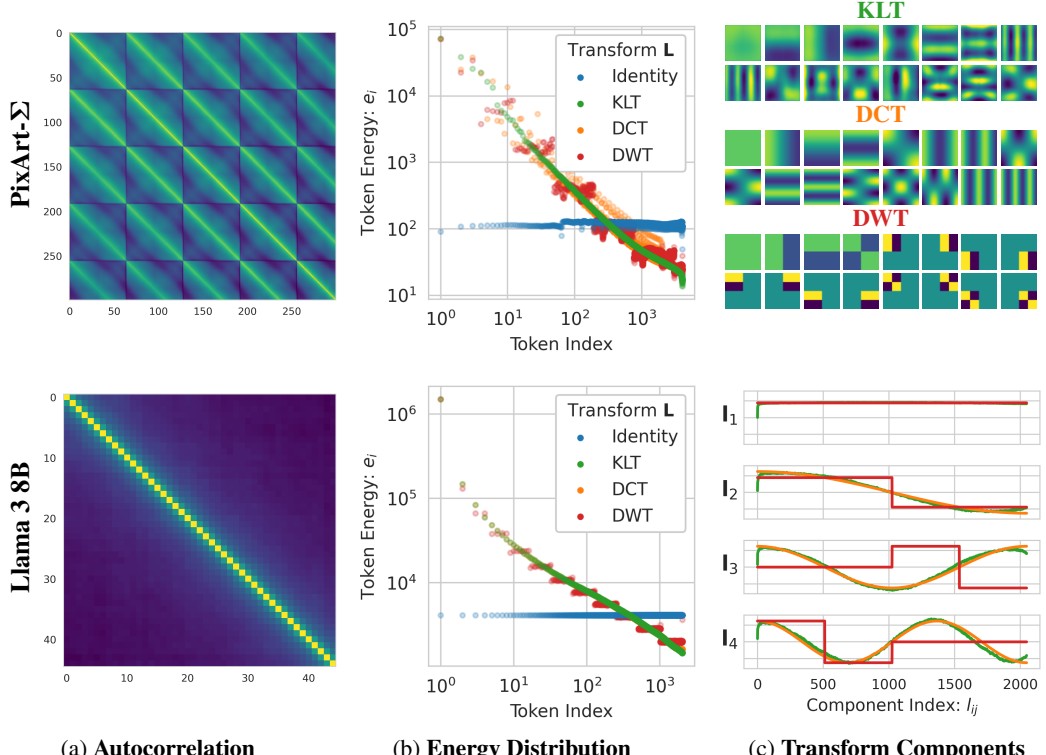

(a) **Autocorrelation**    (b) **Energy Distribution**    (c) **Transform Components**

Figure 3: Visualization of (3a) a portion of the autocorrelation matrix, (3b) transformed token energy distribution, and (3c) transformation components for the input to the cross-attention layer 15 of PixArt-$\Sigma$ and the attention layer 20 of LLaMA-v3-8B, computed on COCO and Wikitext, respectively. The block structure in the LVM activations arises from flattening 2D data into a 1D sequence. Both matrices exhibit a Toeplitz-like diagonal structure, allowing their **KLT** eigenbases to be efficiently approximated by **DCT** (Figure 3c), which concentrates the token energy close to optimal distribution (Figure 3b). The **DWT** closely approximates the optimal energy with discrete levels.

Fortunately, the autocorrelation matrix of common LVM and LLM activations $S$ exhibits a strong structure induced by the properties of natural images and text. Figure 3a shows that tokens corresponding to spatially or sequentially adjacent activations are highly correlated, while distant tokens are weakly correlated. As a result $S$ is approximately (block) Toeplitz, whose eigenvectors can be well-approximated by a Fourier basis[1]. In particular, since the autocorrelation is real and symmetric, we can use a **Discrete Cosine Transform** (DCT) instead of a complex Fourier basis.

The complexity of DCT $\mathcal{O}(ds \log s)$ is lower than a full matrix multiplication since the transformation requires only $\log s$ steps of the Fast Fourier Transform algorithm. Further simplification is possible by retaining only the sign of Fourier coefficients, yielding the **Walsh-Hadamard Transform** (WHT), which approximates DCT while enabling more efficient hardware implementations.

Finally, the **Discrete Wavelet Transform** (DWT)[2] further reduces the computational complexity to $\mathcal{O}(ds)$ while effectively concentrating the energy of the activations at each intermediate step. Each DWT step pushes the energy in the first half (one quarter for 2D signal) of the tokens, requiring $\log s$ steps to fully concentrate the energy. Figure 3 compares energy distributions (3b) and basis components (3c) for KLT, DCT, and DWT on intermediate activations of LVM and LLM architectures.

---

[1]This follows from Szegő's theorem.

[2]We use the Haar wavelet for its simplicity and minimal padding requirements.

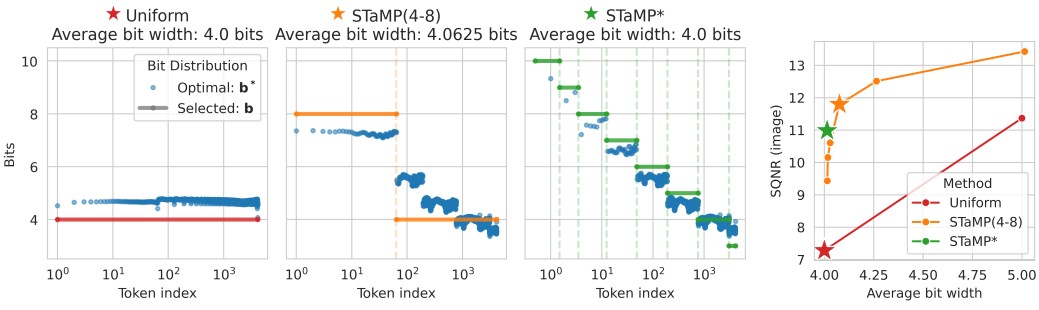

(a) **Bit width allocation and transformation strategies**   (b) **Bit width SQNR tradeoff**

Figure 4: **Comparison of energy and bit width allocation strategies**. Figure 4a compares a uniform allocation without transformation against two STaMP strategies based on the DWT transform. Restricting to only 4- and 8-bits precision (in yellow) introduces minimal overhead while significantly improving output quality. The results in Figure 4b are obtained by varying the number of high-precision tokens and adjusting the uniform quantization bit width for a per token-activation-quantized PixArt-$\Sigma$ model with QuaRot feature transformations.

### 3.3 OPTIMAL BIT ALLOCATION

Given a vector of energies $e$ and a total maximum allocation of $B$ bits, the optimal bit distribution $b$ follows the logarithm of the squared root of the token energy: $b_i^* = \log_2 \sqrt{e_i} - C$, with $C = \left(B - \sum_{i=1}^s \log_2 \sqrt{e_i}\right)/s$. In practice, we are restricted to integer bit widths $\lfloor b^* \rceil$. Furthermore, due to practical limitations, it is beneficial to use only a small number of different bit precisions which are supported on hardware, such as 4 or 8 bits.

For this reason, although the DWT energy concentration is sub-optimal, its property of creating a discrete number of energy levels makes it more suitable to our use case. As illustrated in Figure 4a in yellow, we propose a simple allocation scheme that uses only two bit widths: the first 64 tokens are kept at 8 bits, while the rest uses 4 bits, resulting only in a minor bit width overhead (4.0625 on the PixArt-$\Sigma$), while significantly improving the model accuracy (Figure 4b). For this reason, STaMP with DWT and 2 precision level will be our main focus in the experimental section.

## 4 RELATED WORK

Quantization is a fundamental technique for reducing the computational and memory footprint of deep neural networks, enabling efficient inference with minimal accuracy loss. The rapid growth of LLMs and LVMs has intensified interest in post training quantization (PTQ) (Nagel et al., 2021; Gholami et al., 2022), as retraining these models is often impractical. Recent PTQ approaches focus on removing the outliers and reducing the dynamic range of weights and activations, to improve quantization robustness.

SmoothQuant (Xiao et al., 2023) reduces activations outliers by scaling the feature channels, shifting quantization difficulty from activations to weights. *QuaRot* and *FlatQuant* apply invertible feature transforms over weights and activations to spread outliers across channels, employing randomized Hadamard (Ashkboos et al., 2024) or learning lightweight affine transforms (Sun et al., 2025). van Breugel et al. (2025) develop transforms that commute with transformer operations, such that they can be merged into linear weights. Zhao et al. (2025) develops a Static-Dynamic Channel Balancing (SDCB) procedure based on channel scaling and mixing on Diffusion Transformer (DiT) architectures and retaining certain quantization-sensitive layers to higher bitwidth to achieve W8A8. Li et al. (2025) absorbs activation outliers into a high-precision low-rank branch via singular value decomposition, while quantizing the residuals to 4-bit with a per-token/per-group quantization granularity to achieve W4A4 mixed-precision quantization on DiT. Federici et al. (2025) reduces the dynamic range of DiT activations by subtracting the sequence average from each token and applying Hadamard feature rotations, at the cost of processing an extra token.

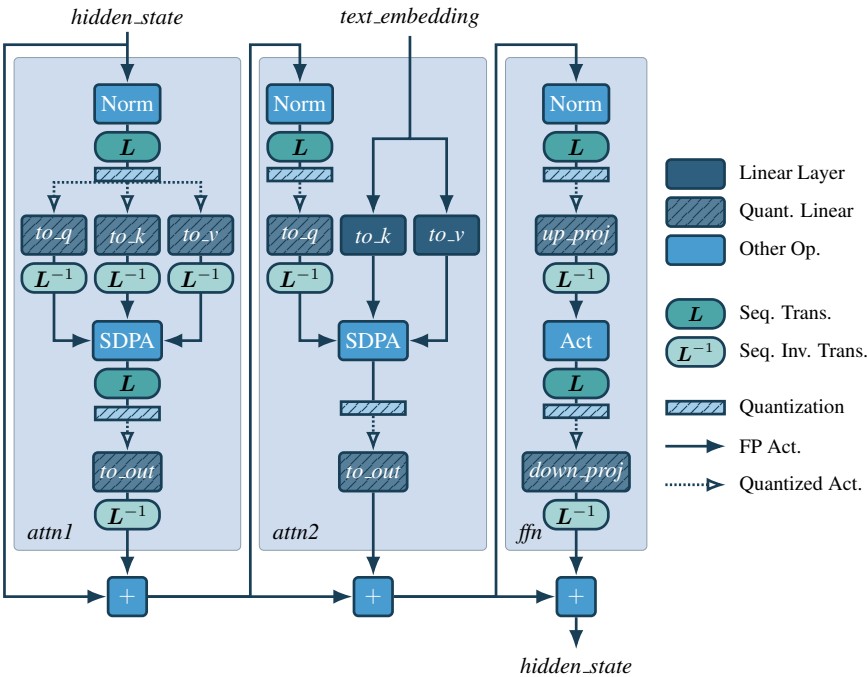

Figure 5: Diagram of a LVM transformer block based on the PixArt-$\Sigma$ architecture. The quantization and sequence transformations operations are indicated explicitly. LLM architectures use the same quantization scheme for the *attn1* and *ffn* blocks. Note that no transform is applied to the *attn2.to_out* activations since the sequence autocorrelation does not present the block diagonal structure because of its dependency on the pooled textual embedding.

While these approaches have advanced PTQ for large models, they operate exclusively along the feature dimension and ignore correlations across the sequence dimension. In contrast, STaMP introduces sequence-aware transformations that exploit local token correlations by using a mixed precision allocation strategy under a fixed average budget. Our approach is strongly relates to classical media compression techniques that leverage frequency-domain transforms to concentrate energy and enable adaptive quantization. JPEG (Leger et al., 1991) and JPEG2000 (Christopoulos et al., 2002) rely on the Discrete Cosine Transform and Discrete Wavelet Transform, respectively, to decorrelate spatial data and allocate bits based on perceptual importance. Similar principles underlie modern video coding standards such AVC (Wiegand et al., 2003) and HEVC (Sullivan et al., 2012), as well as audio codecs like MP3 and AAC, which use the Modified Discrete Cosine Transform (Princen et al., 1987). STaMP demonstrates that it is possible to apply similar principles to the activation space of generative models, where local correlation in the sequence dimension allows energy compaction and selective precision allocation, improving quantization efficiency without retraining.

## 5 EXPERIMENTAL RESULTS

We demonstrate the effectiveness of STaMP on both LVM and LLM models, comparing against uniform activation quantization, and showing its effectiveness in combination with popular feature transformation and weight quantization approaches. Throughout this section, we focus on STaMP with DWT due to its computational efficiency. We refer to Figure 5 for a visualization of how the sequence transformations are applied to the architectures, and to Appendix C for additional results.

### 5.1 VISION LANGUAGE MODELS

**Set-up** Our LVM experiments focus on the DiT architectures of PixArt-$\Sigma$ (Chen et al., 2024b) and SANA (Xie et al., 2025). We quantize activation before each linear layer in transformer blocks using asymmetric quantization with minmax scaling. As common practice, unless otherwise specified, we

Table 1: **STaMP consistently improves LVM quantization.** Image SQNR and Image Reward (IR) for W4A4 per block quantization with block size 64. For all the STaMP results we keep 64 tokens at 8-bits. STaMP consistently improves baselines.

| | | COCO | | | | MJHQ | | | |
| | | SQNR | | IR | | SQNR | | IR | |
| | **STaMP →** | ✗ | ✓ | ✗ | ✓ | ✗ | ✓ | ✗ | ✓ |
|---|---|---|---|---|---|---|---|---|---|
| PixArt-Σ | FP | $+\infty$ | | 0.90 | | $+\infty$ | | 0.96 | |
| | RTN | 5.88 | 6.16 | 0.38 | 0.80 | 5.75 | 6.23 | 0.38 | 0.76 |
| | ViDiT-Q | 7.82 | 6.37 | 0.83 | 0.84 | 7.55 | 8.53 | 0.76 | 0.86 |
| | SVDQuant | 8.78 | 9.72 | 0.90 | 0.91 | 8.83 | 9.75 | 0.86 | 0.89 |
| SANA | FP | $+\infty$ | | 0.87 | | $+\infty$ | | 0.97 | |
| | RTN | 8.63 | 9.32 | 0.89 | 0.91 | 8.56 | 9.40 | 0.95 | 0.99 |
| | ViDiT-Q | 10.03 | 10.74 | 0.89 | 0.86 | 10.04 | 10.69 | 0.96 | 0.97 |
| | SVDQuant | 9.99 | 10.69 | 0.87 | 0.90 | 9.88 | 10.51 | 0.93 | 0.98 |

Figure 6: Visualization of PixArt-Σ sample generation for the results reported in Table 1.

use a separate scale $s_i$ for each token and weight output channel. Following common procedure in literature, in LVMs we do not quantize activations and weights corresponding to the cross-attention *key* and *value* since their effect accounts for less than 5% of the runtime (Li et al., 2025).

We evaluate the fidelity of the quantized LVMs by computing the Signal to Quantized Noise Ratio $\text{SQNR}(\boldsymbol{O}, \tilde{\boldsymbol{O}}) = 10 \log_{10} \left( \|\boldsymbol{O}\|_2^2 / \left\| \boldsymbol{O} - \tilde{\boldsymbol{O}} \right\|_2^2 \right)$ both in the diffusion latent space and image space. We further compute CLIP Score (Hessel et al., 2021) to assess alignment with the textual prompt, CLIP IQA (Wang et al., 2023), and Image Reward (Xu et al., 2023) to evaluate the generated image quality. Following standard procedures, we compute the metrics using 1000 prompts and images from the COCO (Lin et al., 2014) and MJHQ (Li et al., 2024) datasets.

We demonstrate the effectiveness STaMP alone and by combining it with other quantization methods developed in recent literature. We combine STaMP with feature transform methods such as SmoothQuant (Xiao et al., 2023) and QuaRot (Ashkboos et al., 2024), and Static-Dynamic Channel Balancing (SDCB) described in ViDiT-Q (Zhao et al., 2025). We further evaluate STaMP with the mixed precision low-rank weight quantization method described in SVDQuant (Li et al., 2025). We apply STaMP before each linear layer in the Transformer blocks, inverting it right after each linear layer. We use 2-dimensional DWT with 64 8-bit (high precision) tokens unless otherwise specified.

Table 2: **STaMP always improves LLM quantization.** Perplexity (PPL) for W4A4KV4 quantization, using the same setting as (Ashkboos et al., 2024; Sun et al., 2025). We use 64 8-bit tokens for activations and KV-cache for all methods and baselines, even if we do not apply the sequence transform (effectively W4A4.125KV4.125). The STaMP sequence transform improves all baselines.

| STaMP $\rightarrow$ | Llama 3 8B ✗ | Llama 3 8B ✓ | Llama 3.2 1B it ✗ | Llama 3.2 1B it ✓ | Llama 3.2 3B it ✗ | Llama 3.2 3B it ✓ | Qwen 2.5 3B it ✗ | Qwen 2.5 3B it ✓ |
|---|---|---|---|---|---|---|---|---|
| FP | 6.14 | | 13.16 | | 11.27 | | 8.56 | |
| RTN | 668 | 95.3 | 1795 | 700 | 483 | 159 | 99723 | 18767 |
| SmoothQuant | 531 | 93.8 | 883 | 407 | 177 | 88.5 | 66929 | 29063 |
| Quarot | 9.05 | 8.66 | 25.78 | 23.72 | 18.43 | 17.57 | 94.86 | 71.13 |
| FlatQuant | 6.89 | 6.77 | 15.72 | 15.16 | 12.71 | 12.40 | 9.29 | 9.19 |

**Results** Table 1 reports the results obtained by combining STaMP with recent LVM quantization method using the same quantization setting for all the baselines. Both activation and weights are quantized at 4 bits with blocks of size 64, following the setup described in (Li et al., 2025). We observe that STaMP consistently improves upon all the reported metrics on different models and architectures resulting in visually more accurate generations, as shown in Figure 6.

## 5.2 NUMBER OF HIGH PRECISION TOKENS

Figure 4b demonstrates the trade-off between bit width and SQNR observed by changing the number of high-precision tokens in STaMP, while fixing the high and low precision bit widths to 8 and 4 bits respectively. Only activations are quantized to focus the analysis solely on the activation quantization error. We observe a sharp increase in SQNR whenever even a few high precision tokens are introduced. Even in the 5 bits regime, STaMP achieves better performance uniform quantization. Additional comparison with per-block activation quantization can be found in Appendix C.

## 5.3 LARGE LANGUAGE MODELS

**Set-up** We evaluate on language models of different sizes and model classes, including Llama 3 8B (Grattafiori et al., 2024), Llama 3.2 1B and 3B instruct, and Qwen 2.5 3B instruct (Qwen et al., 2025). We take popular *feature* transforms from literature, including SmoothQuant (Xiao et al., 2023), QuaRot (Ashkboos et al., 2024), and the state-of-the-art FlatQuant (Sun et al., 2025), and evaluate whether STaMP (DWT) brings additional gains to these baselines. We use the exact same quantization set-up as used in (Ashkboos et al., 2024; Sun et al., 2025), W4A4KV4 per token activation quantization, and like them, evaluate Wikitext-2 (Merity et al., 2017) perplexity at sequence length 2048. We use round-to-nearest (RTN) for weight quantization (vs. GPTQ (Frantar et al., 2022)), as weight quantization is completely perpendicular to sequence transforms. In this setup, we keep the first 64 tokens in 8 bits for all baselines, which means all methods use an effective activation/KV bit width of 4.125 bits. Note that, on LLMs, STaMP can be effectively applied only for the prompt-processing phase, since during token generation, only one activation is available at the time. Despite this limitation, STaMP remains useful for prompt processing, which is typically compute-bound (Agrawal et al., 2023; Chen et al., 2024a; Kamath et al., 2025), as reducing the activation size lowers total compute and latency. See Appendix B.2 for additional details.

**Results** We observe (Table 2) that all baselines improve consistently when STaMP is added. This is especially apparent for scenarios where baselines are far from FP performance—e.g. for the small Llama 3.2 1B and 3B instruct models, which are evidently hard to quantize at 4 bit. This demonstrates that STaMP is not a competitor of other quantization techniques (e.g. feature transforms), but an additional tool for achieving extremely low bit width quantization—which can be added without any manual tuning or training to existing quantization methods. Appendix C includes additional evaluations on few-shot reasoning tasks, which show the same trend as the perplexity evaluation, with STaMP consistently increasing accuracy.

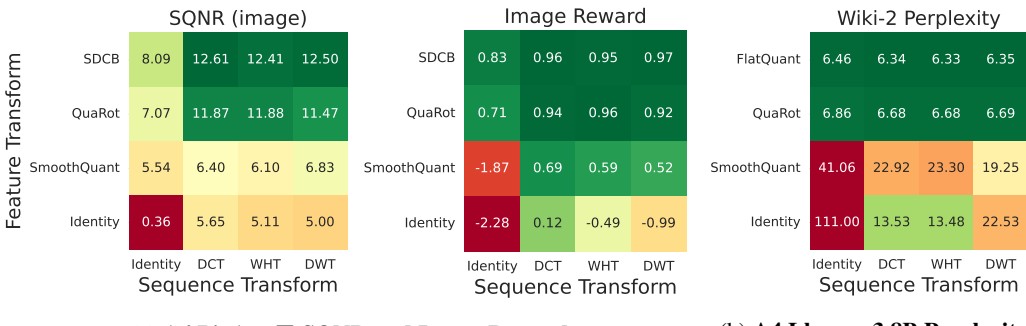

(a) **A4 PixArt-Σ SQNR and Image Reward**

(b) **A4 Llama v3 8B Perplexity**

Figure 7: Effect of combining Feature Transforms (rows) and STaMP (columns) with A4 on the PixArt-Σ and LLama v3 8B models.

## 5.4 COMBINING FEATURE AND SEQUENCE TRANSFORMATIONS

In Figure 7 we assess the effectivenes of STaMP for three different sequence transformations in combination with popular feature transforms. Overall, we observe that improvements are largely complementary, especially for LVMs. Furthermore, results show that DCT, WHT and DWT perform similarly to each other, demonstrating that there is little price to pay for switching from the more accurate and computationally expensive DCT to the cheaper approximate DWT.

## 5.5 OVERHEAD ESTIMATES

Table 3 reports that the theoretical compute overhead and latency of CUDA for a latent denoising step with STaMP (DWT) is comparable to Hadamard transforms on the features, accounting for less than 5% of the total runtime. The DWT transform is applied in three levels, consistently with the results reported in Table 1 and Figure 4a, using a specialized CUDA kernel that considers the memory layout of the activation tensors. Appendix B.3 includes further details on the benchmarking procedure. The minimal overhead in the number of floating point operations (flops) suggests that the latency overhead can be further reduced with better optimized kernels or specialized hardware.

Table 3: Overhead in terms of latency on CUDA and extra flops for a single PixArt-Σ denoising step. STaMP with DWT has little impact on the latency and number of floating point operations.

| Transformation | | Overhead [%] | |
|---|---|---|---|
| Feature | Sequence | flops | CUDA |
| Hadamard | - | 0.24 | 3.0 |
| - | Hadamard | 0.69 | 56.8 |
| - | DWT | 0.21 | 4.8 |
| Hadamard | DWT | 0.44 | 7.8 |

## 6 CONCLUSION

This work explores uncharted territories in the recent advancements of LLM and LVM quantization, applying invertible transformations over the sequence dimension to further reduce quantization error. Drawing inspiration from traditional signal processing, we introduce a novel method that exploits the autocorrelation structure that is naturally present in the intermediate activations of generative models to enable a more efficient mixed-precision activation quantization scheme.

We demonstrate that STaMP complements existing PTQ techniques such as *SmoothQuant* and *QuaRot*, and plays a critical role in advancing low-precision activation quantization, pushing the boundaries of W4A4 quantization for both LLMs and LVMs and offering training-free solution for deploying high-performance models in resource-constrained environments.

ACKNOWLEDGMENTS

Images were generated using the PixArt-$\Sigma$ model licensed under Apache 2.0 and SANA model licensed under Apache 2.0.

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

# A PROOFS

## A.1 PROOF FOR THEOREM 1

We prove the result reported in Theorem 1 in three steps. First, we demonstrate that the quantization error for an orthogonal sequence transformation is the same as the quantization error on the sequence transformed inputs:

$$\mathcal{L}(\boldsymbol{X}; \boldsymbol{L}) = \mathcal{L}(\boldsymbol{LX}) \tag{10}$$

*Proof.* This simply follows from the invariance of the Frobenious norm to orthogonal transformations.

$$
\begin{aligned}
\mathcal{L}(\boldsymbol{X}; \boldsymbol{L}) &= \left\| \boldsymbol{L}^{-1}\mathcal{Q}(\boldsymbol{LX}) - \boldsymbol{X} \right\|_2^2 \\
&= \left\| \boldsymbol{L}^{-1}\mathcal{Q}(\boldsymbol{LX}) - \boldsymbol{L}^{-1}\boldsymbol{LX} \right\|_2^2 \\
&= \left\| \boldsymbol{L}^{-1}(\mathcal{Q}(\boldsymbol{LX}) - \boldsymbol{LX}) \right\|_2^2 \\
&= \left\| \mathcal{Q}(\boldsymbol{LX}) - \boldsymbol{LX} \right\|_2^2 \\
&= \mathcal{L}(\boldsymbol{LX})
\end{aligned}
$$

$\square$

Secondly, for completeness, we prove the upper-bound reported in Equation 3. For quantization scheme with minmax scales shared across all feature channels $s_{ij} = \overline{s}_i$:

$$\mathcal{L}(\boldsymbol{x}_i) \leq \frac{d}{4} \frac{\mathbb{E}\left[\text{range}(\boldsymbol{x}_i)^2\right]}{(2^{b_i} - 1)^2} \tag{11}$$

*Proof.*

$$
\begin{aligned}
\mathcal{L}(\boldsymbol{x}_i) &= \mathbb{E}\left[\left\| \mathcal{Q}(\boldsymbol{x}_i) - \boldsymbol{x}_i \right\|_2^2\right] \\
&= \mathbb{E}\left[\left\| \mathbf{Q}^{-1}\left(\left\lfloor \frac{x_i}{\overline{s}_i} \right\rceil + z_i\right) - \boldsymbol{x}_i \right\|_2^2\right] \\
&= \mathbb{E}\left[\left\| \left(\left\lfloor \frac{\boldsymbol{x}_i}{\overline{s}_i} \right\rceil + z_i - z_i\right)\overline{s}_i - \boldsymbol{x}_i \right\|_2^2\right] \\
&= \mathbb{E}\left[\left\| \left(\left\lfloor \frac{\boldsymbol{x}_i}{\overline{s}_i} \right\rceil - \frac{\boldsymbol{x}_i}{\overline{s}_i}\right)\overline{s}_i \right\|_2^2\right] \\
&= \mathbb{E}\left[\left\| \left(\left\lfloor \frac{\boldsymbol{x}_i}{\overline{s}_i} \right\rceil - \frac{\boldsymbol{x}_i}{\overline{s}_i}\right) \right\|_2^2 \overline{s}_i^2\right] \\
&\leq \mathbb{E}\left[\left\| \frac{\mathbf{1}}{2} \right\|_2^2 \overline{s}_i^2\right] \\
&= \frac{d}{4} \frac{\mathbb{E}\left[\text{range}(\boldsymbol{x}_i)^2\right]}{(2^{b_i} - 1)^2}.
\end{aligned}
$$

$\square$

Lastly, we bound the range using the norm 2:

$$\text{range}(\boldsymbol{x}_i)^2 \leq 2\left\|\boldsymbol{x}_i\right\|_2^2. \tag{12}$$

Equality is attained whenever $\hat{\boldsymbol{x}}_i$ consists of two non-zero entries $-v$ and $v$, for which $\text{range}(\boldsymbol{x}) = 2v$ and $\|\boldsymbol{x}\|_2 = \sqrt{2}v$.

Using this three steps we can write the proof for Theorem 1

*Proof.*

$$\mathcal{L}\left(\boldsymbol{X};\boldsymbol{L}\right) \overset{10}{=} \mathcal{L}\left(\boldsymbol{L}\boldsymbol{X}\right)$$

$$= \sum_{i=1}^{s} \mathcal{L}\left(\boldsymbol{l}_i\boldsymbol{X}\right)$$

$$\overset{11}{\leq} \frac{d}{4} \sum_{i=1}^{s} \frac{\mathbb{E}\left[\text{range}\left(\boldsymbol{l}_i\boldsymbol{X}\right)^2\right]}{(2^{b_i}-1)^2}$$

$$\overset{12}{\leq} \frac{d}{2} \sum_{i=1}^{s} \frac{\mathbb{E}\left[\|\boldsymbol{l}_i\boldsymbol{X}\|_2^2\right]}{(2^{b_i}-1)^2}$$

$\square$

## A.2  OPTIMAL BIT WIDTH ALLOCATION

We determine the optimal bit width allocation $\boldsymbol{b}^*$ for an energy vector $\boldsymbol{e}$ by determining the bit width for which the ratio $e_i/2^{2b_i^*} \approx e_i/(2^{b_i}-1)^2$ is constant for all tokens:

$$\begin{cases} \frac{e_i}{2^{2b_i^*}} = k \\ \sum_{i=1}^{s} b_i^* = B \end{cases} \tag{13}$$

$$\implies b_i^* = \frac{\log_2 e_i - \log_2 k}{2} \tag{14}$$

$$\implies \sum_{i=1}^{s} \frac{\log_2 e_i - \log_2 k}{2} = B \tag{15}$$

$$\implies \log_2 k = \frac{1}{s}\sum_{i=1}^{s} \log_2 e_i - \frac{2B}{s} \tag{16}$$

$$\implies b_i^* = \frac{\log_2 e_i - \frac{1}{s}\sum_{i=1}^{s} \log_2 e_i + \frac{2B}{s}}{2} \tag{17}$$

$$\implies b_i^* = \log_2 \sqrt{e_i} + \underbrace{\frac{B - \sum_{i=1}^{s} \log_2 \sqrt{e_i}}{s}}_{C} \tag{18}$$

## A.3  EFFECTIVENESS OF ENERGY CONCENTRATION

We demonstrate the effectiveness of the proposed Sequence Transform and Mixed precision scheme by considering the upper bound in Equation 8 in two different settings. In order to simplify computation, we will consider $\frac{e_i}{2^{2b_i}}$ instead of $\frac{e_i}{(2^{b_i}-1)^2}$ since the difference between the two quantities is negligible for practical values of $b_i$.

1. **Uniform Energy**:
   In this scenario we have $e_i = E/s$ and $b_i = B/s$. We note that the energy for each token is equal to the average of the squared eigenvalues $\lambda_i^2$ of $\boldsymbol{S}$:

$$e_i = E/s = \text{Trace}(\boldsymbol{S})/s = \frac{1}{s}\sum_{i=1}^{s} \lambda_i^2 \overset{def}{=} \overline{\lambda^2}. \tag{19}$$

   Therefore:

$$\frac{d}{2}\sum_{i=1}^{s} \frac{e_i}{2^{2b_i}} = \frac{d}{2}\sum_{i=1}^{s} \frac{\overline{\lambda^2}}{2^{2B/s}} = \frac{ds}{2}2^{\log_2 \overline{\lambda^2} - 2B/s} \tag{20}$$

2. **Maximum Energy Concentration**:
   We consider a scenario in which the energy corresponds with the squared eigenvalues $e_i =$

$\lambda_i^2$. The optimal bit allocation is given by Equation 18:

$$b_i^* = \log \lambda_i + \frac{B}{s} - \underbrace{\frac{\sum_{i=1}^s \log_2 \lambda_i}{s}}_{\overline{\log_2 \lambda}} \tag{21}$$

Therefore:

$$\frac{d}{2} \sum_{i=1}^s \frac{e_i}{2^{2b_i}} = \frac{d}{2} \sum_{i=1}^s \frac{\lambda_i^2}{2^{\log \lambda_i^2 + \frac{2B}{s} - 2\overline{\log_2 \lambda}}} = \frac{ds}{2} 2^{\overline{\log_2 \lambda^2} - 2B/s} \tag{22}$$

Comparing the two results is equivalent to comparing $\log_2 \overline{\lambda^2}$ (uniform) and $\overline{\log_2 \lambda^2}$ (max concentration). Using Jensen's inequality and the convexity of $log_2$, we have:

$$\overline{\log_2 \lambda^2} = \mathbb{E}[\log_2 \lambda_i^2] \le \log \mathbb{E}[\lambda_i^2] = \log_2 \overline{\lambda_i^2}. \tag{23}$$

Therefore, the Maximum Energy concentration strategies achieves a lower value than the uniform scheme.

## B EXPERIMENTAL DETAILS

### B.1 LVMs

**Baselines** We implement SmoothQuant, QuaRot, ViDiT-Q and SVDQuant closely following the details reported in the paper and the respective codebases. Specifically, for ViDiT-Q (Zhao et al., 2025) we use $\alpha = 0.01$, as reported in their PixArt-$\Sigma$ setup. For SVDQuant and SmoothQuant we use a default value of $\alpha = 0.5$.

**Quantization** In order to promote a fair comparison with the other models, for the results reported in Table 1, instead of using the 8-bits weight quatization scheme described in ViDiT-Q (8 bits for the FFN blocks, first and last transformer blocks), we use per-block weight and activation quantization, as described in SVDQuant (Li et al., 2025), which improves the ViDiT-Q results. In line with the SVDQuant methodology, we retain the depth-wise convolutions within the feed-forward layers of the SANA transformer blocks in full precision. Meanwhile, the two point-wise convolutions are quantized by treating them as linear layers. Notably, unlike the original SVDQuant implementation, our experiments do not incorporate GPTQ at any stage. The activation-quantization only results reported in Figure 4b and Figure 7 use 4 bits per token quantization.

**STaMP** For the STaMP results, excluding the specific ablation study, we use 64 high-precision tokens, which results in an effective activation bit-width of 4.0625 bits on the PixArt-$\Sigma$ model and 4.125 bits on the SANA model. Non-STaMP result do not use any mixed precision tokens.

**SQNR** Our experimental results report the average value of SQNR in decibels (dB) for each image. In other words, for $N$ 2-dimensional signals $\boldsymbol{O}_i, \ldots, \boldsymbol{O}_N$ and their quantized counterparts $\tilde{\boldsymbol{O}}_i, \ldots, \tilde{\boldsymbol{O}}_N$ consisting of $t$ tokens (pixels) and $f$ features, we compute:

$$SQNR = \frac{10}{N} \sum_{i=1}^{N} \log_{10} \frac{\sum_{t=1}^{T} \sum_{f=1}^{F} o_{itf}^2}{\sum_{t=1}^{T} \sum_{f=1}^{F} (o_{itf} - \tilde{o}_{itf})^2} = \frac{10}{N} \sum_{i=1}^{N} \log_{10} \frac{\|\boldsymbol{O}_i\|_2^2}{\|\boldsymbol{O}_i - \tilde{\boldsymbol{O}}_i\|_2^2}.$$

The SQNR of each image is averaged in log space to mitigate the effect of extreme outliers.

### B.2 LLMs

**Quantization** We use dynamic quantization for the KV cache and activations. Quantization scales and offsets are determined per token, sequence, and (for KV) head. We follow (Ashkboos et al., 2024; Sun et al., 2025) and only quantize weights, KV cache, and inputs to linear layers within the transformer block. We use round-to-nearest weight quantization, as weights are unaffected by STaMP—more advanced weight quantization schemes could improve results further, but this is perpendicular to our contribution. we range set the weights by computing the weight quantization squared error for a grid of candidate ranges and selecting the candidate with lowest error. For fairness, for all experiments (incl. baselines) we keep the first 64 tokens in 8 bits, which gives an effective bit width of 4.125 for both STaMP and non-STaMP results.

**Baselines** For SmoothQuant, we calibrate the scales based on the Wikitext-2 training dataset activations. For QuaRot, we follow the original paper and reduce the minmax activation range by 10%. For FlatQuant, we use their recommended settings for all models (e.g. training for 15 epochs on 128 Wikitext-2 training sequences).

**STaMP** The first token in most LLMs acts as an attention sink Xiao et al. (2024); Gu et al. (2025), and typically contains massive outliers Sun et al. (2024). Keeping the first token in 8 bits helps to accurately represent these massive outliers. To ensure the massive outlier stays in the first token, STaMP is not applied to the first token. Note that the baselines that do not use STaMP's transform, do benefit from keeping the first token (and next 63) in high-precision in our experiments.

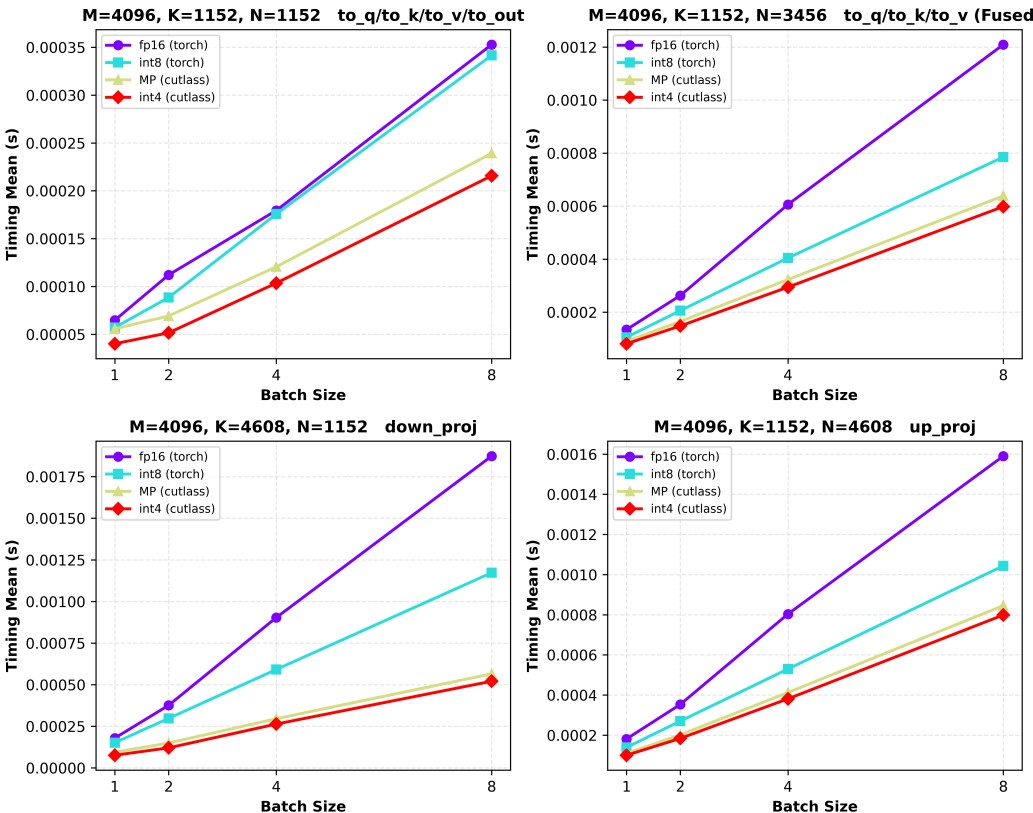

Figure 8: Visualization of runtime estimation in terms of latency on CUDA for running matrix multiplication kernels with different data types, shapes, and batch sizes averaged over 1000 runs. The proposed mixed precision approach has a minimal overhead that does not depend on the batch size.

### B.3 NOTES ON THE RUNTIME ESTIMATION

#### B.3.1 DWT OVERHEAD

To compute the overhead associated with the sequence and feature transforms, we measured the time required to run a single denoising step with the original PixArt-$\Sigma$ model, and compared it with the modified architectures (having the extra transform operations). Measurements are performed on an A100-SXM4-80GB GPU using python 3.10.12, PyTorch 2.5.1+cu121, and CUDA drivers 12.5. Hadamard transforms are based on the CUDA-accelerated kernels from the fast-hadamard-transform package [3]. For Haar DWT operations, we built a specialized CUDA kernel, optimized for applying DWT over the sequence dimension. The increased latency measured with the Hadamard applied on the sequence dimension can be mainly attributed to the memory reshaping operations required to use the fast-hadamard-transform kernel.

#### B.3.2 MIXED-PRECISION MATRIX MULTIPLICATION

When using STaMP, the first tokens of the activation tensors are stored at a higher bit width with respect to the subsequent ones (8 and 4 bits in our experimental settings). This implies that the subsequent linear operation should perform part of the computation with higher precision. The number of extra bit operations for the 8 bit part is minimal (about $1.56\%$ on PixArt-Sigma).

It is important to note that STaMP 4-8 bits mixed precision scheme drastically reduces the number of binary operations when compared to *int8* matrix multiplications. However, ß the latency measure-

---

[3]https://github.com/Dao-AILab/fast-hadamard-transform

ments are heavily influenced by the support for specific data types and operations in the hardware and software stack of the target device. Demonstrating the effectiveness of *int4* quantization is an important step towards developing support for more power-efficient solutions in commercial-grade hardware.

We assessed the effectiveness of the proposed mixed precision design by measuring the time to perform a mixed precision matrix multiplication on CUDA, and we compared to performing the same operation with *fp16*, *int8* or *int4* data types. When using Nvidia hardware, one can take advantage of Tensor Cores to accelerate low-precision operations and allow higher computational throughput. PyTorch uses Tensor Cores kernels for most of the low-precision data types (such as *tf32*, *fp16*, *bf16*, *int8*), but the PyTorch stack lacks an *int4* accelerated kernel. Following the procedure from Sun et al. (2025) [4], we used a Cutlass-based *gemm* kernel with minor modifications to the python bindings to enable inplace operations.

In Figure 8, we report the outcome of our benchmakrs for different batch sizes and for the tensor shapes used in the Pixart-$\Sigma$ architecture. The measured overhead for the mixed-precision matrix multiplication over the *int4* one for batch-size 2 (required for generating a single output image) vary from about 30% (for smaller shapes) to approximately 10% (for fused Q/K/V computation and up projection). Increasing the batch size to 8 reduces the relative overhead to under 10% on all layers.

One should consider that our Mixed-Precision implementation essentially runs two matmuls: one using the custom cutlass kernel over the *int4* data double-packed into an *int8* tensor, the other using the standard torch *int8* kernel over the first 64 tokens and writing the results on the first 64 tokens of the output tensor. For batch size 2, we also perform an additional *int8* operation over the first 64 tokens of the second sample. For batch sizes 4 and larger we slice the tensors to retrieve the first 64 tokens of each batch element and pack into contiguous memory before running a single *int8* matmul. Additional memory transfer optimization and kernel fusion could further decrease the overhead introduced by the STaMP operations.

## C  ADDITIONAL RESULTS

### C.1  LVMs

We report additional results for the setups described in Section 5, including an ablation of the effect of STaMP on different activations (Table 4), bit width vs SQNR comparison against per-block activation quantization (Figure 10), and additional metrics (Table 5) and visualizations (Figure 11) for the results reported in the main text.

Table 4: Effect of A4 activation quantization on single activations of the Pixart-$\Sigma$ model on the Image SQNR. Each entry corresponds to a model for which only the corresponding input activation is quantized. Note that STaMP has little to no effect on *attn2.to_out* since the structure is determined by pooled textual embedding, which does not present the same correlation structure as the other activations in the network.

| Transform | *attn1* | *attn1.to_out* | *attn2.to_q* | *attn2.to_out* | *ffn.up_proj* | *ffn.down_proj* |
|---|---|---|---|---|---|---|
| Identity | 6.01 | 10.09 | 0.40 | 15.48 | 1.15 | 4.92 |
| QuaRot | 13.80 | 16.01 | 16.29 | 23.20 | 11.36 | 8.42 |
| STaMP | 7.68 | 13.89 | 11.60 | 15.49 | 8.75 | 9.61 |
| QuaRot+STaMP | 15.28 | 16.78 | 18.31 | 23.23 | 13.84 | 14.91 |

### C.2  LLMs

We report additional LLM evaluation involving common few-shots reasoning tasks in Table 6. The evaluation is performed using the *lm-eval-harness* toolkit (Gao et al., 2024).

---

[4]https://github.com/ruikangliu/FlatQuant

Figure 9: Visualization of the Images generated with A4 activation quantization and several combination of Feature and Sequence transforms for the Pixart-$\Sigma$ architecture. The images refer to the same setup described in Figure 7.

## D   USE OF LLMS

Large Language Models have been used only for sentence rephrasing and grammar check, which resulted in minor alterations to the paper text.

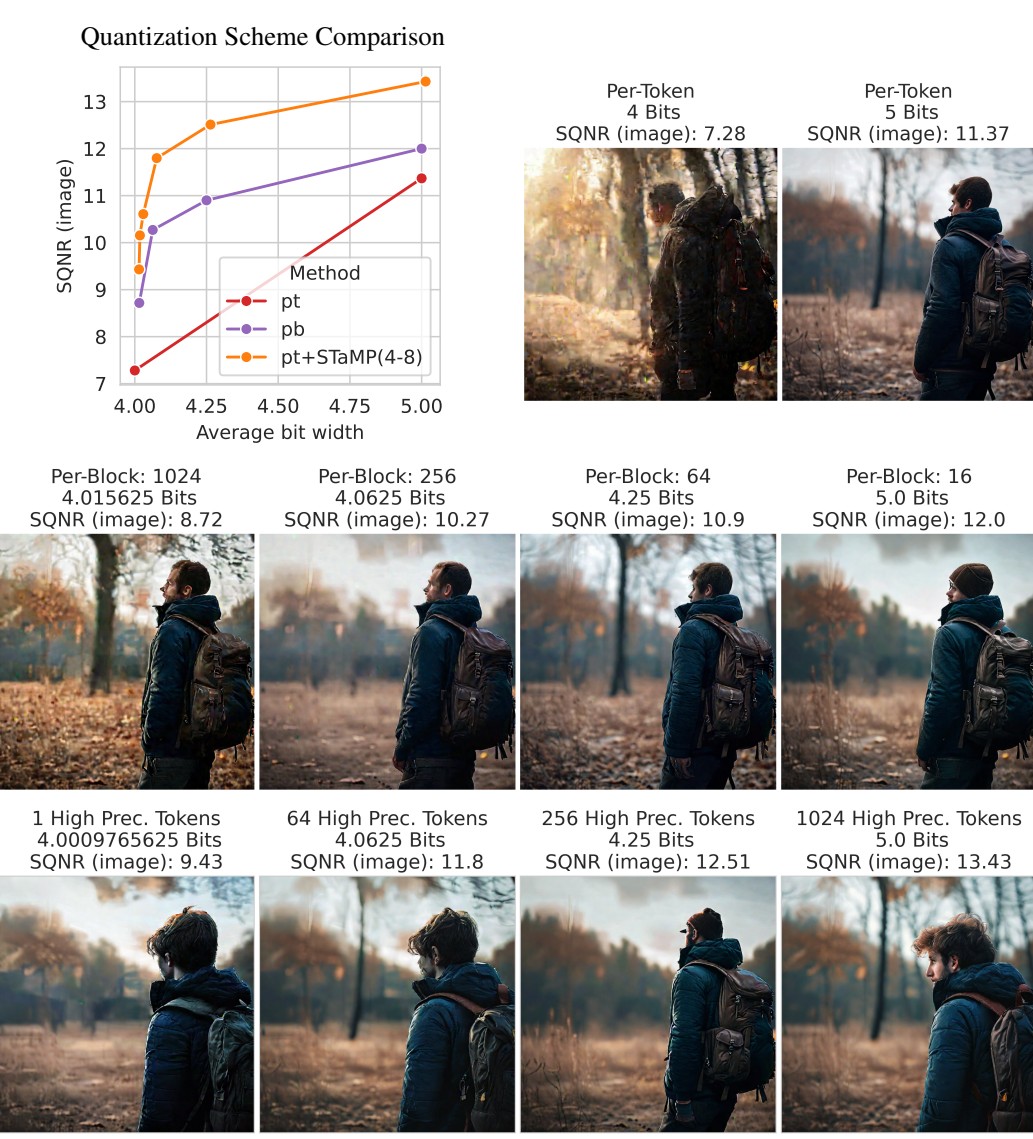

Figure 10: Tradeoff between SQNR and bit width for per-token activation quantization (pt) per-block activation quantization (pb) at different block size (from 16 to 1024), and per-token with STaMP (pb+STaMP) at varying number of high precision tokens on the PixArt-$\Sigma$ model. We consider 16 bits scales for each scale parameter. The visualization correspond to the Prompt *'A guy with a backpack looking at the ground to his left.'*.

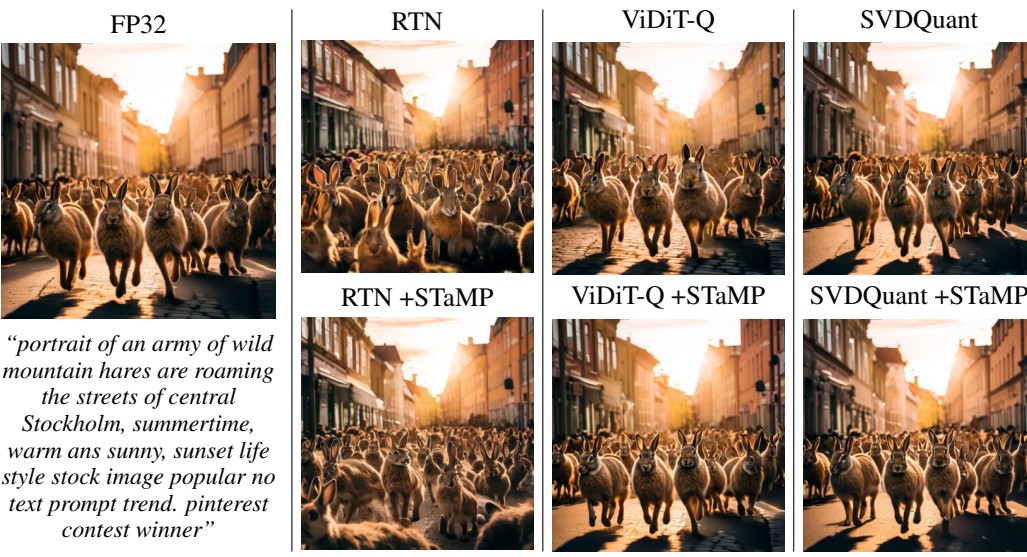

Figure 11: Visualization of a sample generation for the SANA for the results reported in Table 1.

Table 5: Additional metrics for the experiments reported in Table 1.

| Dataset | LVM | Method | STaMP | CLIP | CLIP IQA | SQNR (latent) |
|---------|-----|--------|-------|------|----------|---------------|
| COCO | PixArt-Σ | FP | | 31.54 | 0.91 | $+\infty$ |
| | | RTN | ✗ | 31.23 | 0.79 | 0.19 |
| | | | ✓ | 31.88 | 0.91 | 0.49 |
| | | SVDQuant | ✗ | 31.68 | 0.91 | 0.96 |
| | | | ✓ | 31.76 | 0.91 | 1.47 |
| | | ViDiT-Q | ✗ | 31.63 | 0.87 | 0.73 |
| | | | ✓ | 31.92 | 0.87 | 1.03 |
| | SANA | FP | | 31.85 | 0.89 | $+\infty$ |
| | | RTN | ✗ | 31.85 | 0.89 | 3.39 |
| | | | ✓ | 31.94 | 0.89 | 4.09 |
| | | SVDQuant | ✗ | 31.83 | 0.90 | 4.61 |
| | | | ✓ | 31.88 | 0.90 | 5.30 |
| | | ViDiT-Q | ✗ | 31.79 | 0.89 | 4.63 |
| | | | ✓ | 31.82 | 0.89 | 5.42 |
| MJHQ | PixArt-Σ | FP | | 31.46 | 0.85 | $+\infty$ |
| | | RTN | ✗ | 29.90 | 0.75 | 0.28 |
| | | | ✓ | 30.96 | 0.87 | 0.71 |
| | | SVDQuant | ✗ | 31.13 | 0.86 | 1.27 |
| | | | ✓ | 31.28 | 0.87 | 1.86 |
| | | ViDiT-Q | ✗ | 30.97 | 0.81 | 0.86 |
| | | | ✓ | 31.20 | 0.83 | 1.31 |
| | SANA | FP | | 31.57 | 0.83 | $+\infty$ |
| | | RTN | ✗ | 31.51 | 0.84 | 4.45 |
| | | | ✓ | 31.60 | 0.84 | 5.39 |
| | | SVDQuant | ✗ | 31.52 | 0.83 | 5.69 |
| | | | ✓ | 31.55 | 0.83 | 6.46 |
| | | ViDiT-Q | ✗ | 31.60 | 0.83 | 5.76 |
| | | | ✓ | 31.60 | 0.83 | 6.65 |

Table 6: LLM evaluation on Common Reasoning tasks. The experimental setup is equivalent to the one described in Table 2.

| Task | STaMP → | Llama 3 8B ✗ | Llama 3 8B ✓ | Llama 3.2 1B it ✗ | Llama 3.2 1B it ✓ | Llama 3.2 3B it ✗ | Llama 3.2 3B it ✓ | Qwen 2.5 3B it ✗ | Qwen 2.5 3B it ✓ |
|---|---|---|---|---|---|---|---|---|---|
| *arc challenge* | FP | 54.4 | | 36.1 | | 46.4 | | 56.0 | |
| | RTN | 43.9 | 44.1 | 27.9 | 28.4 | 35.7 | 35.9 | 36.6 | 37.0 |
| | SmoothQuant | 40.3 | 40.2 | 30.5 | 30.5 | 34.5 | 34.6 | 39.2 | 39.3 |
| | QuaRot | 39.0 | 38.7 | 30.5 | 30.5 | 37.4 | 37.1 | 42.8 | 43.2 |
| | FlatQuant | 52.0 | 52.5 | 33.9 | 34.2 | 44.4 | 45.5 | 52.0 | 54.0 |
| *arc easy* | FP | 84.2 | | 70.7 | | 79.7 | | 83.5 | |
| | RTN | 75.2 | 75.8 | 56.2 | 56.3 | 63.6 | 63.8 | 66.7 | 66.8 |
| | SmoothQuant | 71.4 | 72.1 | 61.7 | 62.4 | 64.5 | 65.1 | 69.0 | 69.3 |
| | QuaRot | 71.5 | 71.3 | 58.2 | 58.3 | 66.5 | 66.5 | 73.5 | 73.6 |
| | FlatQuant | 81.3 | 82.4 | 67.6 | 67.8 | 77.5 | 77.7 | 82.2 | 82.4 |
| *hellaswag* | FP | 61.3 | | 44.3 | | 52.8 | | 56.2 | |
| | RTN | 39.2 | 50.2 | 31.0 | 34.0 | 37.2 | 41.9 | 32.4 | 33.2 |
| | SmoothQuant | 39.1 | 47.4 | 33.3 | 36.5 | 40.1 | 45.3 | 33.3 | 33.8 |
| | QuaRot | 49.5 | 51.2 | 37.9 | 38.9 | 45.9 | 46.9 | 34.0 | 33.9 |
| | FlatQuant | 58.7 | 59.7 | 41.2 | 41.9 | 50.4 | 50.8 | 53.6 | 54.5 |
| *lambada* | FP | 71.1 | | 55.1 | | 63.3 | | 59.3 | |
| | RTN | 0.2 | 1.2 | 0.3 | 0.9 | 0.7 | 3.6 | 0.0 | 0.0 |
| | SmoothQuant | 0.5 | 2.7 | 0.6 | 1.6 | 2.4 | 9.0 | 0.0 | 0.0 |
| | QuaRot | 15.1 | 24.6 | 16.4 | 20.6 | 37.6 | 40.6 | 0.0 | 0.0 |
| | FlatQuant | 67.1 | 67.1 | 44.1 | 48.3 | 58.1 | 58.9 | 55.0 | 57.5 |
| *piqa* | FP | 81.1 | | 74.0 | | 77.3 | | 78.0 | |
| | RTN | 75.4 | 76.5 | 66.2 | 66.5 | 69.7 | 70.3 | 71.5 | 71.6 |
| | SmoothQuant | 74.1 | 74.3 | 68.7 | 68.8 | 71.6 | 72.0 | 72.1 | 72.3 |
| | QuaRot | 74.2 | 73.9 | 68.9 | 68.4 | 70.9 | 70.7 | 73.1 | 72.9 |
| | FlatQuant | 79.4 | 80.2 | 72.1 | 73.0 | 76.1 | 75.7 | 77.5 | 77.6 |
| *winogrande* | FP | 77.5 | | 62.0 | | 70.6 | | 70.6 | |
| | RTN | 71.0 | 71.0 | 55.2 | 55.2 | 63.1 | 63.1 | 62.0 | 62.0 |
| | SmoothQuant | 70.8 | 70.8 | 54.9 | 54.9 | 64.8 | 64.8 | 65.6 | 65.6 |
| | QuaRot | 70.8 | 70.8 | 57.6 | 57.6 | 62.6 | 62.6 | 65.9 | 65.9 |
| | FlatQuant | 75.2 | 74.9 | 58.6 | 60.1 | 66.4 | 68.8 | 66.6 | 68.1 |
| Average | FP | 69.7 | | 55.7 | | 63.8 | | 65.5 | |
| | RTN | 50.8 | 53.1 | 39.4 | 40.2 | 45.0 | 46.5 | 44.9 | 45.1 |
| | SmoothQuant | 49.4 | 51.2 | 41.6 | 42.5 | 46.3 | 48.5 | 46.5 | 46.7 |
| | QuaRot | 53.3 | 55.1 | 44.9 | 45.7 | 53.5 | 54.1 | 48.2 | 48.2 |
| | FlatQuant | 67.4 | 67.9 | 50.9 | 52.3 | 60.7 | 61.5 | 62.7 | 63.9 |

