# OpenReview forum: "STaMP: Sequence Transformation and Mixed Precision for Low-Precision Activation Quantization"
_ICLR.cc/2026/Conference — ICLR 2026 Poster_

### Official Review · Reviewer_jGRZ · 2025-10-23

**Soundness:** 3
**Presentation:** 3
**Contribution:** 3
**Rating:** 4
**Confidence:** 4

**Summary:**

This paper proposes STaMP (Sequence Transformation and Mixed Precision), a novel post-training activation quantization method for LLMs and LVMs. Unlike prior work that applies invertible feature transformations across the channel dimension, STaMP introduces linear sequence transformations to exploit correlations between neighboring tokens. By concentrating activation energy into a few tokens and assigning them higher precision under a fixed average bit budget, STaMP effectively reduces quantization error. The authors approximate the optimal Karhunen–Loève Transform using efficient DCT or DWT implementations and derive theoretical bounds on quantization loss. Extensive experiments on PixArt-Σ, SANA, and LLaMA/Qwen models show consistent improvements over existing methods such as SmoothQuant, QuaRot, and SVDQuant, both in SQNR and downstream performance, with minimal computational overhead.

**Strengths:**

1. The paper introduces sequence-domain transformations for activation quantization, a direction largely unexplored compared to prior channel-wise or feature-domain approaches. It can be combined with existing quantization methods and feature transformations without retraining or altering model weights.
2. Extensive results on both LVMs (PixArt-Σ, SANA) and LLMs (LLaMA 3, Qwen 2.5) show clear improvements in SQNR, perplexity, and visual quality under low-bit settings (W4A4), confirming robustness and generality.
3. The paper provides clear figures (e.g., Figs. 2–6) illustrating how energy concentration improves quantization performance and complements feature transforms.

**Weaknesses:**

1. The paper does not quantify the additional latency or memory cost of applying and inverting sequence transforms in large-scale models. Without such analysis, it is difficult to assess whether the efficiency gains from lower-bit activations outweigh the added transform cost in practical deployment scenarios.
2. For the decoding stage of LLMs, STaMP may not be directly applicable, since each newly generated token would require recomputing the sequence transform matrix L.

**Questions:**

1. How would STaMP operate during autoregressive decoding, where tokens are generated sequentially? Since the sequence transform matrix L require access to the full sequence, does it mean that we need to recompute L in every decoding step?
2. Could you provide quantitative measurements of the runtime and memory overhead introduced by applying and inverting the sequence transform in large transformer models? How does this cost compare to the savings from reduced activation bitwidth?
3. How is the mix-precision processed in STaMP? Specifically, since there are 64 tokens kept at 8 bits, and the rest uses 4 bits, how do you run the mix-precision model efficiently?

---

> ### Author Response · Authors · 2025-11-21
>
> We first want to thank the reviewer for recognizing the novelty of our method, its clarity and for praising our extensive evaluation.
> Addressing the reviewer concerns:
>
> ## STaMP Overhead and Mixed Precision Kernel
> The latency measurements strongly depend on hardware characteristic and software support. However, we acknowledge the importance of quantifying the impact of STaMP in terms of compute and latency on common hardware. We report these measurements on our shared response to the reviewers, which is also included in the paper. Regarding the memory, DWT does not require additional memory allocation for inference. Additionally, our mixed precision activation use only 1.5% more memory than INT4 activation (or 49% less than INT8).
>
> The implementation of the mixed precision kernel strongly depends on the specific hardware. In our evaluation we split the tensor separating the (contiguous) activations that require 8 bits from the rest, run the int4 and int8 kernel in parallel and the re-insert the activations back in the tensor after de-quantization. An alternative implementation could involve bit packing/unpacking: the 64 8-bits activations can be mapped into 128 4-bits activations. This operation would be followed by a pure int4 matmul kernel and a packing operation involving sums and simple bit-shifts. We did not further explore this venue since the main bottleneck on current deep-learning on CUDA hardware is related to the lack of int4 native support. Overall, we want to further underline that our method results in a significant reduction of the total number of arithmetic operations.
>
> ## LLM token generation
> The reviewer is correct in their assessment, on LLMs, STaMP is relevant solely during the prompt processing phase since the other activations are not kept in memory during token generation. Since the prompt-processing is usually compute-bound \[1,2,3\] (especially for mobile devices), the use of STaMP activations can reduce the total compute (and therefore latency). During token generation, on the other hand, the token rate is mostly dictated by the latency of memory transfer \[1,2,3\]. In this setting, aggressive (online) activation quantization methods such as STaMP are less relevant since linear layer matmuls involve only 1 token at the time. We clarified this crucial aspect in the LLM-specific portion of the experimental section (lines 274-478).
>
> ### References
> [1] Agrawal, Amey, et al. "Sarathi: Efficient llm inference by piggybacking decodes with chunked prefills." arXiv preprint arXiv:2308.16369 (2023).
>
> [2] Chen, Hao Mark, et al. "Progressive mixed-precision decoding for efficient llm inference." arXiv preprint arXiv:2410.13461 (2024).
>
> [3] Kamath, Aditya K., et al. "Pod-attention: Unlocking full prefill-decode overlap for faster llm inference." Proceedings of the 30th ACM International Conference on Architectural Support for Programming Languages and Operating Systems, Volume 2. 2025.

---

### Official Review · Reviewer_sLp3 · 2025-10-30

**Soundness:** 3
**Presentation:** 4
**Contribution:** 4
**Rating:** 6
**Confidence:** 4

**Summary:**

This manuscript introduces STaMP (Sequence Transformation and Mixed Precision), a novel post-training quantization (PTQ) framework aimed at addressing low-bit activation quantization in large generative models (LLMs and LVMs). The core innovation, which distinguishes this work from prior art focused on feature-dimension transforms, is the application of linear transformations (e.g., DWT) along the sequence dimension. This method seeks to exploit local token correlations to compact activation energy into a minority of transformed tokens. Subsequently, the framework employs a mixed-precision strategy, allocating higher bit-widths to these high-energy tokens while quantizing the remainder more aggressively. The authors demonstrate through experiments that STaMP can serve as an orthogonal module that  consistently improves the performance of various existing PTQ methods under a W4A4 setting, without requiring any retraining.

**Strengths:**

1. High Novelty: The primary strength of this work lies in its high degree of originality. By shifting the focus of quantization transforms from the feature dimension to the sequence dimension, the authors introduce a new research paradigm.
 2. Solid Experimental Results: The paper's claims are supported by consistent and significant performance improvements across a range of models (LLMs and LVMs), datasets, and strong baselines.
3. Excellent Complementarity: The experiments clearly demonstrate that STaMP is an orthogonal improvement that can be combined with state-of-the-art methods like QuaRot to achieve further gains.
4. Clear Motivation and Exposition: The analogy to classical media compression provides a strong theoretical basis and an intuitive understanding of the proposed method.

**Weaknesses:**

1. Unaddressed Implementation Overhead: The most significant weakness is the failure to discuss or quantify the runtime overhead of the per-token mixed-precision scheme on current hardware. The lack of native hardware support may necessitate inefficient dequantization operations, which would severely impact the method's practical speedup.
2. Unquantified Transform Cost: While the computational cost of the DWT/DCT transform is theoretically small, the paper fails to provide a simple quantitative analysis to substantiate this claim (e.g., by comparing it to the computational load of an FFN layer in a Transformer block), leaving its relative overhead ambiguous.

**Questions:**

1. Regarding practical implementation on current mainstream GPUs (e.g., NVIDIA A100/H100), could the authors elaborate on a viable strategy for the proposed mixed-precision scheme? What is the anticipated latency overhead introduced by dequantization or conditional logic compared to a uniform W4A4 implementation?
2. To contextualize the cost of the proposed transform, could the authors provide a brief computational complexity analysis comparing the $O(ds)$ cost of DWT to the $O(s \cdot d \cdot d_{ffn})$ cost of a typical Transformer FFN layer, thereby quantifying the relative overhead introduced by the transform itself?
3. In the LLM experiments, a fixed strategy of keeping the first 64 tokens in high precision was applied to all methods. Have the authors considered whether this is an optimal strategy for the baseline methods, which do not employ sequence transforms? An ablation study that implements and compares against a dynamic, magnitude-based token selection strategy for the baselines could further strengthen the fairness of the experimental comparison.
4. To strengthen the paper's motivation, particularly the analogy to media compression, further insight is requested. Technologies like JPEG use energy compaction to more aggressively compress high-frequency information, which is less perceptible to the human eye. What is the conceptual equivalent in the context of STaMP? What kind of information corresponds to the low-energy components after the DWT/DCT transform, which are subsequently assigned lower precision? A more intuitive explanation of which aspects of the token sequence are deemed "less critical" by the transform would significantly enhance the persuasiveness of the paper's motivation.
$\textbf{If these issues can be resolved, I will consider giving a higher score.}$

---

> ### Author Response · Authors · 2025-11-21
>
> Thanking the reviewer for recognizing the novelty, clarity, applicability of our method and the relevance of our experimental section. To address their comments and questions.
>
> ## STaMP Overhead
> We include a detail analysis of the overhead of the mixed precision scheme and DWT in our shared answer. Thanking the reviewer for highlighting this limitation, we include this analysis in the main text to complement our experimental evaluation. Our analysis includes both an estimate of the computational costs and assessment on CUDA hardware. Our INT4 matmul latency evaluations are currently hindered by current lack of torch support for int4 matmuls rather than a fundamental limitation of our method. In summary, we expect our latency to be quite close to a pure int4 kernel with a limited transformation overhead ( DWT +  inverse ~ FHT).
>
> ## Sorting By Magnitude Baseline
> As validated in other work in the literature, our experimental results have shown that LLM output is mostly sensitive to the precision of the first token. Figure 3b indicates that the average magnitude of the following tokens does not noticeably change, hinting to the limited effectiveness of keeping more tokens in high-precision. The reviewer’s proposed dynamic strategy seems quite interesting, although it would incur the extra cost of computing the magnitudes and (partially) sorting activations online, which would probably be comparable to DWT in cost. The proposed method also requires keeping the ordering in memory for the inverse transformation.
>
> We performed experiments for the proposed setup and report the wikitext-2 perplexity for RTN in the following table in which the requested baseline is reported as “sort by $|x|$” . For completeness, we consider sorting activations by the norm 2 and their absolute maximum. In all the experiments, the first 64 tokens for each activations are kept at 8 bits
>
> | | Llama 3 8B | Llama 3.2 1B it | Llama 3.2 3B it | Qwen 2.5 3B |
> |---|----|---|---|---|
> | No Sorting |  668 |1795 | 483 | 99723 |
> | Sort by $\|x\|_2$ | 693 | 1884| 513| 100623|
> | Sort by $\|x\|_\infty$ |686 | 1892| 490 |103385|
> | STaMP | 95.3 | 700 | 159 |18767 |
>
> Overall, the proposed baseline seems to perform slightly worse than simply quantizing the first 64 tokens at higher precision. We speculate this is due to 2 main factors:
> 1. The magnitude (norm 2) of the tokens that are input to most of the layers is similar or essentially the same due to the numerous LayerNorm blocks. This can also be observed in Figures 2B and 3b. We also empirically observe that sorting by the maximum achieves only marginally better results that sorting by norm 2.
> 2. Errors in the first tokens can propagate to the following ones due to the autoregressive nature of attention layers. This suggests that if the magnitude of the token is similar, we are probably better off if we keep the token that occurs first in higher precision.
>
> ## High frequency tokens
> The low-energy tokens after applying DWT correspond to the high-frequency signal in the (compressed) latent space. In LVMs, the signal still retain the spatial characteristics, with the difference that 1 latent token token corresponds to multiple (16x16) pixels in the image space. Intuitively, the consideration regarding the limited importance of high-frequency signal is analogous to the one we can make for standard image compression, with the difference that high-frequency components in the latent space correspond to medium-high frequency in the image space.
> Empirically, we found that completely removing high-frequency components from the latent space, or applying more aggressive quantization to them, introduces visible checkerboard artifacts in the resulting images.
>
> In LLMs, we can speculate that the high frequency signal may be interpreted as localized changes that relate to synonyms and local choices of words rather than long-range dependencies that correlate with the meaning or structure of the entire sentence.
>
> Ultimately, our choice to preserve low-frequency tokens at higher precision is driven by their larger quantization scale, rather than any perceptual criteria, which are not straightforwardly mapped to activation space due to the non-linear network interactions.

---

> ### Comment · Reviewer_sLp3 · 2025-11-26
>
> Thank you for your response.
>
> However, I note that the int4 and int8 kernels you utilized are not sufficiently state-of-the-art. In fact, adequately optimized int4 kernels, often achieved through CUTLASS programming, have been available within the community for some time, and you haven't actually employed these for deployment.
>
> Given this lack of real-world deployment (which is crucial for quantization methods), I will maintain my score of 6.

---

> > ### Author Response · Authors · 2025-11-27
> >
> > We sincerely thank the reviewer for engaging in the discussion and for pointing us to the CUTLASS int4 kernel, which helped us strengthen our performance evaluation.
> >
> > Our primary contribution is a novel methodology for improving activation quantization, validated across diverse tasks and models. While kernel-level optimization is highly hardware-specific and was not the central focus of this work, we appreciate the suggestion and agree that assessing efficiency with optimized kernels adds practical relevance.
> >
> > To address this, we integrated the CUTLASS int4 kernel from \[1\] and conducted an extensive benchmark study. The updated measurements are:
> >
> > | Data Type | Kernel | bops [10^12] | CUDA [ms] |
> > |---|---|---|---|
> > | fp16 |Torch | 2.199 | 0.615 |
> > | int8 | Torch | 1.099 | 0.354 |
> > | int4 | CUTLASS | 0.550 | 0.176 |
> > | int8[64]+int4 | CUTLASS | 0.558 |0.196 |
> >
> > These results confirm that STaMP's mixed-precision scheme incurs only ~10% overhead compared to pure int4 CUTLASS kernels, while delivering over 40% speedup compared to Torch int8. We included extensive benchmarking with mixed precision across several matmul shapes and batch sizes in Appendix B.3.2 to demonstrate consistent scalability and efficiency gains.
> >
> > We hope the reviewer agrees that these findings strengthen the practical applicability of our method.
> >
> >
> >
> > ### Reference
> > \[1\] Sun, Yuxuan, et al. FlatQuant: Flatness Matters for LLM Quantization. arXiv, 2024.

---

### Official Review · Reviewer_jfMb · 2025-10-30

**Soundness:** 4
**Presentation:** 3
**Contribution:** 3
**Rating:** 6
**Confidence:** 5

**Summary:**

This paper proposes Sequence Transformation and Mixed Precision(STaMP), a new PTQ method that operates along the sequence dimension, different from most recent methods that focus on utilizing rotation to modify along the feature dimension. The key idea is to exploit local correlation between sequential tokens by applying a sequence-wise linear transform (e.g., DCT, DWT) before quantization, followed by a mixed-precision strategy that assigns higher bit-widths to a subset of tokens carrying higher "energy."

**Strengths:**

1. New Perspective: The paper introduces quantization transformations along the sequence dimension—a novel and orthogonal approach compared to existing methods. This direction effectively exploits temporal or spatial token correlations often overlooked by prior research.

2. Solid Theoretical Analysis: The work offers a well-defined mathematical treatment of quantization error and establishes a theoretical upper bound (Theorem 1) that connects token energy, bit allocation, and resulting error.

3. Synergistic with Existing Methods: The proposed approach integrates seamlessly with leading quantization strategies (e.g., QuaRot, SmoothQuant, FlatQuant, SVDQuant), delivering complementary and cumulative performance improvements.

**Weaknesses:**

1. Missing Real-Time and Latency Evaluation: The paper does not report inference speed, latency, or hardware efficiency metrics—critical aspects for assessing quantization performance. Given that the multiplication of XL occurs online, it would be valuable to analyze its impact during inference.

2. Absence of Calibration Set Ablation: There is no investigation into how the size of the calibration dataset influences performance, which is an essential factor for post-training quantization (PTQ) methods.

**Questions:**

See Above

---

> ### Author Response · Authors · 2025-11-21
>
> We thank the reviewer for recognizing the novelty of our work and for praising our theoretical analysis and applicability of our method. We address the reviewer’s comments hereafter.
>
> ## Inference Latency
> Evaluating the real-time latency of our method requires addressing a number of hardware-specific nuances, which are not the main goal of our submission. However, we recognize the importance of assessing the latency and compute requirements of our method, which we extensively address in our shared answer. To summarize, sequence DWT (+ inverse) results in a comparable overhead to well-established approaches such as Fast Hadamard Transform (QuaRot) on features.
>
> ## Calibration set Ablation
> Concerning the ablation on the calibration set size, we want to underline that, similar to QuaRot, STaMP is a parameter-free approach that does not require calibration. The only hyper-parameter concerns the number of high-precision tokens, which is fixed to 64 because of data-independent considerations regarding the activation size and compute overhead. Only the baselines including a SmoothQuant (and FlatQuant for LLMs) component require a scale calibration, which is performed consistently to the description reported in the corresponding papers. Therefore, any investigation regarding calibration dataset size would primarily impact the baselines rather than STaMP, given its parameter-free nature.

---

### Official Review · Reviewer_F9G6 · 2025-10-31

**Soundness:** 3
**Presentation:** 4
**Contribution:** 2
**Rating:** 4
**Confidence:** 4

**Summary:**

1. The paper proposes STaMP quantization to mitigate accuracy loss in low-precision activation quantization for LLMs and LVMs.
2. Unlike existing feature-wise methods, STaMP applies linear transformations along the sequence dimension to exploit local correlations in language and visual data.
3. It uses a mixed-precision scheme: a small number of high-energy tokens are kept at higher precision to maintain accuracy with lower average bit-widths.
4. STaMP complements established feature transformations and weight quantization methods without conflicting with them.
5. Experiments on models like Llama 3 and PixArt-Σ demonstrate consistent accuracy improvements in 4-bit activation quantization.
6. For efficiency, STaMP adopts practical transforms instead of complex KLT, balancing energy concentration and computational cost.

**Strengths:**

1. The paper is really well-written in both figures, tabs, equations, and texts.
2. The paper contains experiments on both SD and LLM.
3. The method design features strong complementarity and is compatible with the existing technical system.

**Weaknesses:**

1. I a m confused  with the argument in QuaRot that rotation only occurs in the feature dimension, because some weights (such as up/gate projection) are left-multiplied by the rotation matrix.
2. The experiments on Large Language Models is insufficient: (1) The model size is relatively small, and it has not been scaled up to the magnitude greater than 8B; (2) There are only PPL experiments, with the lack of mainstream zero-shot reasoning experiments, which are more crucial for verifying the effectiveness of PTQ.
3. The paper introduces left-invertible matrices, but it does not provide a detailed and clear explanation of how such matrices are integrated into the weight parameters of the original architecture (like QuaRot and SpinQuant).
4. As a work focusing on quantization, the paper does not report experimental results related to accelerated inference.

**Questions:**

See Weakness.

---

> ### Author Response · Authors · 2025-11-21
>
> We thank the reviewer for acknowledging the clarity, applicability of our method, and broad coverage of our experimental evaluations. In response to the reviewers comments we want to highlight a few details regarding our method.
>
> ## Clarification on ”Left” transforms
> The reviewer is correct in pointing out that previous work applies left weight transformations. However, in our work, contrarily to previous literature, we consider activation as matrices with shape [sequence, features], and refer to **left** transformations as matrix multiplication applied to the sequence dimension of the activations. This operation differs substantially to the operation applied in QuaRot and SpinQuant since we are “mixing” across different tokens instead of transforming different channel for each token separately, as previous work. For this reason, our transformations cannot be fused into the matrix weights, and we need to invert them right after the linear layers (Equation 7). In order to clarify how STaMP is applied to LLM and LVM architecture we added a new figure (Figure 5) describing how the transformations are applied and inverted in the self-attention, cross-attention and FFN layers, including the quantization operations.
>
> ## On LLM evaluation
> The main goal of this paper is to demonstrate the effectiveness of STaMP on a wide range of models (LLMs and LVMs) and architectures. We don’t expect a fundamental difference in performance arising when increasing the LLM size, therefore, we focused on covering multiple architectures rather than using our compute on a few large LLMs.
>
> However, we recognize the importance of including zero/few-shot evaluations to further quantify the effect of STaMP. Therefore, we additionally evaluate the models on Common Sense Reasoning tasks (arc_easy, arc_challenge, hellaswag, lambada, piqa, winogrande).
> Due to limitation on markdown tables, here we report the average task accuracy w/o and with STaMP (in parenthesis) for the 4 LLMs, using the same quantization setup used to produce Table 2. A full table including detailed results for each separate task is included in the supplementary material (Table 7).
>
> | Method |  Llama 3 8B | Llama 3.2 1B it | Llama 3.2 3B it | Qwen 2.5 3B it |
> |------|---------|----------|-------|-------|
> | RTN | 50.8 (53.1) | 39.4 (40.2) | 45.0 (46.5) | 44.9 (45.1) |
> | SmoothQuant | 49.4 (51.2) | 41.6 (42.5) | 46.3 (48.5) | 46.5 (46.7) |
> | QuaRot | 53.3 (55.1) | 44.9 (45.7) | 53.5 (54.1) | 48.2 (48.2) |
> | FlatQuant | 67.4 (67.9) | 50.9 (52.3) | 60.7 (61.5) | 62.7 (63.9) |
>
> Overall, the results of this evaluation are consistent with the observed perplexity measurement: STaMP reliably improves performance when compared to the baselines.
>
> ## Inference runtime
> Thanking the reviewer for the relevance of their comment regarding the inference timing, we refer to our shared response for a detailed analysis of latency and compute for STaMP transforms and mixed precision scheme. These results are now included in the manuscript. To summarize, STaMP adds little overhead compared to a pure int4 kernel. Our transformations ( DWT forward+inverse) introduce similar latency to existing solutions (e.g. QuaRot).

---

### Author Response · Authors · 2025-11-21
**Shared Response**

We thank the reviewers for the insightful comments and questions which provided valuable feedback to improve our submission; and for recognizing the novelty, clarity and applicability of STaMP. We hereby address the shared concern about lack of assessment of the overhead for the proposed transform and mixed precision scheme. First, we want to underline that the real-time performance is heavily influenced by many hardware-related factors (e.g. presence of Tensor cores) and software support for some specific operation (int4/int8 matmuls), which vary greatly depending on the platform (e.g. desktop vs mobile). For this reason, other than relying solely on CUDA latency, we also highlight on number of floating point (flops) and binary operations (bops), which are indicative of the overall number of required arithmetic operations. We articulate our analysis in two parts covering the performance of the DWT and mixed precision scheme, respectively. Recognizing the relevance of these aspects, results are also included in the main text of the paper (Section 5.5, Appendix B.3).


To summarize, STaMP (with DWT) introduces an overhead comparable to other methods in literature (such as Fast Hadamard Transform) with little overhead compared to vanilla A4W4. The latency measurement, in practice, are heavily influenced by the support for some specific operations (such as int4 matmuls). Nevertheless, we believe that demonstrating the applicability of more aggressive activation quantization methods can drive the demand for specific hardware/software solutions to further improve on the energy and latency for the inference process of large generative models.

---

> ### Author Response · Authors · 2025-11-21
> **DWT overhead**
>
> First, we want to highlight that DWT requires a few sequential operations consisting of simple sums and subtractions, which can be represented as convolutions. Each layer is applied to all feature channels simultaneously and an exponentially smaller portion of the input sequence (E.g. 3 levels $s+s/4+s/16 = 21/16 s$ on LVMs).
>
> We implemented a custom CUDA DWT kernel to assess the overhead of applying DWT (and its inverse) online for each linear layer.
> The number in the reported table are based on 10 full denoising forward steps for the PixArt-Sigma architecture compared to forward passes without any transform, run on an Nvidia A100-SXM4-80GB with CUDA 12.5, python 3.10.12 and torch 2.5.1+cu121. Overall, the overhead of DWT (forward + inverse) on sequence is comparable to applying one online Hadamard transform on feature channels using the fast Hadamard transform (FHT) \[1\].
>
> | Feature Transform | Sequence Transform | flops Overhead [%] | CUDA Overhead [%] |
> |---------|----------|-------|------|
> | FHT     | -        | 0.24  | 1.65 |
> | -     | FHT        | 0.69  | 53.9 |
> | -     | DWT      | 0.21  | 3.49 |
> | FHT     | DWT      | 0.44 | 5.07 |
>
> Note that naively applying the FHT on the sequence dimension, would result in significantly more overhead. This is also due to the memory re-shaping required to apply FHT on the sequence dimension. Furthermore, DWT on sequence requires slightly less flops than Hadamard transform on features accounting for about 0.2% of the operations of the full model. The use of a more optimized kernel or specialized hardware (such as FPGA, currently used for DWT/DCT for real-time image and video encoding/decoding) could further reduce the impact of DWT.
>
>
> ### References
> \[1\]: https://github.com/Dao-AILab/fast-hadamard-transform

---

> ### Author Response · Authors · 2025-11-21
> **Mixed Precision Kernel overhead**
>
> We assessed the effectiveness of our proposed mixed precision design by measuring the number of bit operations (bops) and timing on CUDA hardware for matmuls using fp16, int8, int4, and our mixed-precision (int8\[64\]+int4) data types. When using an Nvidia hardware to accelerate low-precision operations one should use Nvidia Tensor Cores to allow higher throughput. Torch already provides support for accelerated fp16 and int8, but does not support int4 due to the experimental nature of int4 on Ampere architecture. Given the scarce support for int4 on Nvidia, we use a custom matmul kernel \[2\] targeting Ampere for our benchmarks. We benchmark a matrix multiplication operation for an activation having 4096 tokens and 4096 feature channels, which is similar to the shape we observe in LLMs and LVMs. Additional results for larger matmuls are included in Table 4 in the Appendix.
>
>
>
>
>
> | Data Type        | Kernel | bops [10^12] | CUDA [ms] |
> |-------------------|--------|--------|--------|
> | fp16             | torch  | 2.199  | 0.615  |
> | int8             | torch   | 1.099  | 0.354  |
> | int8             | custom | 1.099  | 0.617  |
> | int4             | custom | 0.550  | 0.382  |
> | int8[64] + int4  | custom | 0.558  | 0.405  |
>
>
>
> We want to highlight that the overhead in terms of number of bops introduced by the mixed precision setup is minimal, since only 1.5 % of the tokens are in 8-bits precision. The overhead for int4 and for the mixed precision on CUDA, when compared to torch int8/fp16, is mostly due to the lack of proper int4 support on the CUDA/torch stack, while the custom int4 kernel has a substantial speedup when compared to a modification of the same kernel optimized to run with int8. Compared to the int4 matmul, the mixed precision overhead on CUDA is around 6%.
>
>
>
> It should be considered that we are merely running two matmuls splitting the input tensor and merging the two results, so this percentage could be lower with an optimized mixed-precision kernel implementation (e.g. optimizing the weights memory transfer from device to SMs shared memory). Notably, the number of additional bops required for DWT and its inverse is minimal (about 0.2% for most matmuls in PixArt-sigma linear layers) since the operation is linear in the number of tokens.
>
> ### References
> \[2\]: https://github.com/carsonpo/quadmul

---

> ### Author Response · Authors · 2025-11-27
>
> We thank Reviewer sLp3 for highlighting the optimized int4 kernel. Following this suggestion, we updated our benchmarks to include a CUTLASS-based int4 implementation (as in \[3\]), enabling us to demonstrate practical speedups on Nvidia hardware compared to int8. The updated results are shown below, and a detailed analysis of our mixed precision setting across various matrix shapes and batch sizes is provided in Appendix B.3.2:
>
> | Data Type | Kernel | bops [10^12] | CUDA [ms] |
> |---|---|---|---|
> | fp16 |Torch | 2.199 | 0.615 |
> | int8 | Torch | 1.099 | 0.354 |
> | int4 | CUTLASS | 0.550 | 0.176 |
> | int8[64]+int4 | CUTLASS | 0.558 |0.196 |
>
> Overall, our mixed-precision kernel incurs only ~10% overhead compared to a pure int4 kernel, while achieving 1.8 speedup over int8 matmuls. This overhead further decreases for larger matrices and batch sizes.
> We believe these additional results strengthen the practical relevance of our approach and complement its scientific contribution.
>
> ### Reference
> \[3\] Sun, Yuxuan, et al. FlatQuant: Flatness Matters for LLM Quantization. arXiv, 2024.

---

### Meta-Review · Area_Chair_ByYV · 2025-12-29

**Summary:**

This paper introduces a new activation quantization mechanism for LLMs and LVMs. Unlike existing outlier reduction methods that apply linear transformations along the hidden dimension, the proposed approach operates along the sequence dimension. Overall, I find this idea novel. The authors also did a great job addressing reviewers’ questions during the rebuttal period.

**Reviewer Concerns:**

A main concern raised by multiple reviewers was the lack of latency evaluation, which the authors addressed in their global response. They also added additional experimental results evaluating their method on common benchmarks beyond perplexity. Many other reviewer concerns (e.g., comparisons with quarot and the lack of calibration set ablations) were also clarified during the rebuttal.

However, several issues remain insufficiently addressed. First, the proposed method can only be applied during the prefilling stage (i.e., the prompt processing phase), which is a significant limitation. Many reasoning models produce long responses involving chain-of-thought and self-reflection, which make the decoding stage dominating latency and memory usage. However, the proposed method is unable to be applied in this stage.

Second, the proposed method appears to introduce additional latency overhead compared to pure INT4 quantization.

Finally, existing methods (e.g., quarot) allow linear transformations to be fused into the weight matrices, significantly reducing latency. In contrast, the proposed method does not support such fusion.

**Reviewer Scores:**

Overall, I find the rebuttal to be effective. Based on the discussion above, I would expect that some reviewers may raise their scores after the rebuttal, and that this paper will likely be a borderline acceptance.

---

### Decision · Program_Chairs · 2026-01-26

Accept (Poster)